# Hard Labels In! Rethinking the Role of Hard Labels in Mitigating Local Semantic Drift

**Jiacheng Cui**[1] **Bingkui Tong**[1] **Xinyue Bi**[1] **Xiaohan Zhao**[1] **Jiacheng Liu**[1] **Zhiqiang Shen**[1]

## Abstract

Soft labels from teacher models are a *de facto* practice for knowledge transfer and large-scale dataset distillation (e.g., SRe$^2$L, LPLD). However, when we limit the number of crops per image to reduce the substantial cost of storing precomputed soft labels, these methods suffer severely from *local semantic drift*: visually ambiguous crops can cause soft supervision to deviate from the image-level ground-truth semantics, leading to persistent errors and a train–test distribution mismatch. We revisit the overlooked role of hard labels and show that, when properly integrated, they can act as a content-invariant semantic anchor that calibrates such drift. We theoretically analyze the emergence of drift under sparse soft-label supervision and demonstrate that hybridizing hard and soft labels restores alignment between visual content and semantic supervision. Building on this insight, we propose a new training paradigm, **H**ard Label for **A**lleviating **L**ocal Semantic **D**rift (**HALD**), which uses hard labels as intermediate corrective signals while preserving the fine-grained benefits of soft labels. Extensive experiments on dataset distillation and large-scale classification benchmarks show consistent generalization improvements. On ImageNet-1K, our method achieves 42.7% accuracy with only 285M soft-label storage (reduces by **100$\times$**), outperforming prior state-of-the-art LPLD by 9.0%. Code is available at https://github.com/Jiacheng8/HALD.

## 1. Introduction

Soft labels have emerged as a standard and strong supervision signal derived from pretrained teacher models in knowl-

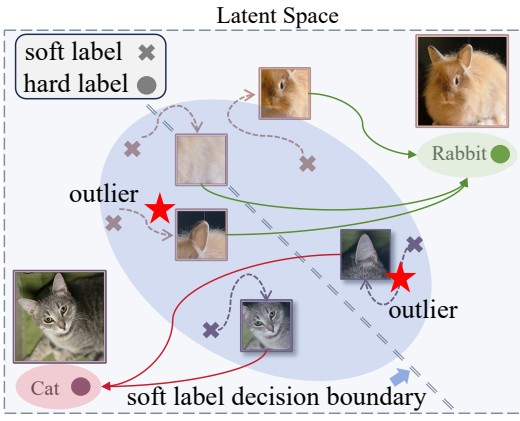

*Figure 1.* Illustration of *local-view semantic drift*: partial crops may change object–label relations, yielding semantics that deviate from the full image.

edge distillation (Hinton et al., 2015) and dataset distillation (Yin et al., 2023) tasks. Unlike hard labels, which provide only class-level supervision, soft labels encode richer inter-class similarity information, offering smoother gradients and better generalization. In particular, for dataset distillation, FKD-based (Shen & Xing, 2022) soft labels have become indispensable because they allow student models to inherit semantic knowledge from powerful teacher networks without relying on access to the teacher in the post stage. This is especially critical in the post-training stage, where the teacher must be completely isolated from the training pipeline as it is trained on the original data, to avoid information leakage and any direct contact with raw full data following the task setting.

Despite these advantages, the reliance on soft labels introduces a critical bottleneck: *storage*. The most widely adopted strategy, such as in FKD and FerKD, involves pre-computing and saving soft labels for every image crop in the distilled dataset. While effective, this design leads to massive storage requirements, particularly on large-scale datasets like ImageNet-1K (*distilled data*: 750 MB vs. *soft-label*: 28.33 GB), even larger than the distilled data storage size which is not acceptable. Thus, storing per-crop logits across thousands of classes results in prohibitive memory overhead, hindering the scalability and practical deployment of dataset distillation pipelines. As datasets continue to grow

---

[1]Department of Machine Learning, MBZUAI, Abu Dhabi, United Arab Emirates. Correspondence to: Zhiqiang Shen <Zhiqiang.Shen@mbzuai.ac.ae>.

*Proceedings of the 43$^{rd}$ International Conference on Machine Learning*, Seoul, South Korea. PMLR 306, 2026. Copyright 2026 by the author(s).

in size and granularity, compressing or reducing soft label storage has become an urgent problem.

A straightforward approach to alleviate storage costs is to reduce the number of crops, and consequently, the number of soft labels per image. However, this seemingly simple solution introduces a more subtle and overlooked issue: *local semantic drift*. As shown in Fig. 1, since crops often capture only partial or ambiguous regions of an image, their soft labels may semantically shift toward unrelated categories. For example, a crop from a cat image can be similar to a rabbit, and the soft embedding derived from the teacher misaligns with the global semantics of the original class. This mismatch between local visual evidence and global semantics undermines training, leading to degraded generalization and unstable predictions. We also provide a theoretical guarantee by establishing a strictly positive lower bound on the expected mismatch between the objective defined with reduced crops and that with sufficient crops. Our analysis shows that this gap is inversely proportional to the number of crops: *the fewer the crops, the larger the mismatch.*

**Hard labels as a corrective signal.** In contrast, hard labels are content-invariant or independent and immune to local visual ambiguity. While they lack the fine-grained information encoded in soft labels, they provide a stable supervisory anchor tied to the ground-truth semantic identity of the image. This raises a key insight: hard labels, if carefully integrated, could serve as corrective signals to calibrate soft-label supervision and mitigate semantic drift. Surprisingly, this potential has been largely overlooked in the literature, where hard labels are often considered too coarse or discarded entirely in favor of soft labels. From a theoretical perspective, we further guarantee that proper joint training with soft and hard labels does not introduce gradient inconsistencies that would hinder optimization. On the contrary, the controlled fluctuations arising from hard-label supervision inject additional information, boosting the learning of new knowledge beyond what soft labels alone can provide. On large-scale ImageNet-1K, our method achieves 42.7% accuracy with only 285M soft-label storage (reduces by **100×**), outperforming prior state-of-the-art LPLD by 9.0%.

**Our contributions.** 1) We revisit the role of hard labels in dataset distillation and introduce a hybrid training paradigm, **H**ard Label for **A**lleviating **L**ocal Semantic **D**rift (**HALD**). 2) Our key idea is to strategically incorporate hard labels to recalibrate the semantic space of image crops while preserving the nuanced information provided by soft labels. 3) We theoretically show that limited soft labels inevitably induce semantic drift and mathematically demonstrate how hard labels mitigate this effect. 4) Extensive experiments across multiple benchmarks confirm that **HALD** reduces distribution mismatch and improves generalization, even under *aggressive soft-label compression*.

## 2. Related Work

**Dataset Distillation.** Dataset distillation aims to construct a small, synthetic surrogate of a large dataset that retains its core information content. The goal is to accelerate training and cut storage costs while achieving performance close to training on the full data. Current approaches can be broadly grouped into six families: 1) *Gradient Matching* (Zhao et al., 2021; Zhao & Bilen, 2021; Lee et al., 2022; Kim et al., 2022; Zhou et al., 2024). 2) *Meta-Model Matching* (Wang et al., 2018; Nguyen et al., 2021; Loo et al., 2022; Zhou et al., 2022; Deng & Russakovsky, 2022; He et al., 2024). 3) *Trajectory Matching* (Cui et al., 2023; Chen et al., 2023; Guo et al., 2024). 4) *Distribution Matching* (Wang et al., 2022; Zhao & Bilen, 2023; Xue et al., 2025; Lee et al., 2022; Sajedi et al., 2023; Shin et al., 2023; Liu et al., 2022). 5) *Decoupled Optimization* (Yin et al., 2023; Shao et al., 2024a;b; Yin & Shen, 2024; Zhang et al., 2025; Cui et al., 2025a; Tran et al., 2025; Shen et al., 2025; Sun et al., 2024). 6) *Diffusion Based* (Gu et al., 2024; Su et al., 2024; Chen et al., 2025; Zhao et al., 2025; Chan-Santiago et al., 2025; Wang et al., 2025; Zou et al., 2025). A comprehensive overview of recent advances can be found in (Liu & Du, 2025; Shang et al., 2025b).

**Soft Label and Hard Label Usage.** Soft labels are widely adopted in dataset distillation for their richer target structure relative to hard labels, enabling finer guidance during optimization (Yin et al., 2023; Qin et al., 2024; Sun et al., 2024; Yu et al., 2025; Cui et al., 2025b). However, storing per-sample soft targets can introduce a substantial memory overhead, often larger than the image storage itself. To mitigate this, LPLD (Xiao & He, 2024) proposes generating a limited set of soft targets and reusing them throughout training, substantially reducing the label-storage budget. (Yu et al., 2025) proposes a label-lightening framework HeLlO that leverages effective image-to-label projectors to generate synthetic labels online from synthetic images. In parallel, GIFT (Shang et al., 2025a) fuses hard information into soft targets to obtain more reliable supervision.

## 3. Approach

**Preliminaries: Dataset Distillation.** Given dataset $\mathcal{O} = \{(x_i, y_i)\}$, dataset distillation seeks a small set $\mathcal{C} = \{(\tilde{x}_j, \tilde{y}_j)\}$ ($|\mathcal{C}| \ll |\mathcal{O}|$) such that models trained on $\mathcal{C}$ and $\mathcal{O}$ generalize similarly:

$$\min_{\mathcal{C}} \sup_{(x,y) \sim \mathcal{O}} |\mathcal{L}(f_{\theta_{\mathcal{O}}}(x), y) - \mathcal{L}(f_{\theta_{\mathcal{C}}}(x), y)|. \quad (1)$$

Here, $\theta_{\mathcal{O}}$ and $\theta_{\mathcal{C}}$ are obtained via ERM (Vapnik, 1991) on $\mathcal{O}$ and $\mathcal{C}$, respectively. With this setup in place, prevailing evaluations of distilled datasets rely on pre-generated soft labels, which tends to underemphasize the role of hard labels, despite their zero storage cost and direct ground-truth supervision. Moreover, storing soft labels (often per

crop/augmentation) can exceed the images themselves, motivating storage-efficient alternatives. We therefore revisit this design choice and analyze the consequences of a soft-only protocol, particularly under limited soft-label coverage.

**Soft Label Recap.** Using a teacher model to generate soft labels (Hinton et al., 2015) for training a new model has become both common and popular, especially in the field of dataset distillation (Wang et al., 2018), where it has repeatedly been shown to be particularly effective for large-scale datasets. The main drawback of soft labels, however, is that each crop requires storing its own soft label, which leads to substantial storage overhead (Yin et al., 2023). A straight-forward workaround is to reduce the number of crops (and thus soft labels) per image (Xiao & He, 2024).

However, we identify an often-overlooked issue that arises when only a small number of soft labels are used per image: *Semantic Shift.* As illustrated in Fig. 1, soft labels are usually assigned to image crops, but these crops may only capture partial regions. This semantic shift problem is intrinsic to soft labels, whereas hard labels, being content-invariant, do not suffer from such drift. While hard labels bring their own limitation: they fail to align the semantic label with the fine-grained visual content, making it difficult for the model to learn detailed representations.

Our work addresses precisely this trade-off. In the following sections, we first provide a theoretical analysis showing why using too few soft labels per image introduces a semantic shift, leading to mismatched train–test distributions and degraded predictions. We then demonstrate how, when used appropriately, hard labels can serve as a corrective signal to calibrate this mismatch, since they provide supervision independent of crop content. Finally, we propose a new training paradigm, *Soft–Hard–Soft*, and show through both theoretical explanation and empirical visualizations that it effectively resolves the limitations of existing approaches.

### 3.1. Training with Limited Soft Label Coverage

**Definition 3.1** (Local-View Semantic Drift). Fix $\tilde{x}$ and augmentation distribution $\mathcal{T}(\tilde{x})$. For $x^{(\text{crop})} \sim \mathcal{T}(\tilde{x})$, let $\tilde{p}(x^{(\text{crop})}) \in \Delta^C$ be the teacher's soft prediction, and define

$$\bar{p} := \mathbb{E}[\tilde{p}(x^{(\text{crop})})], \qquad \Sigma := \text{Cov}[\tilde{p}(x^{(\text{crop})})].$$

For the $s$-crop aggregation $\hat{p}_s := \frac{1}{s} \sum_{i=1}^{s} \tilde{p}(x_i^{(\text{crop})})$, the supervision error decomposes exactly as

$$\mathbb{E}\|\hat{p}_s - e_y\|_2^2 = \underbrace{\|\bar{p} - e_y\|_2^2}_{(\text{I})} + \underbrace{\frac{\text{Tr}(\Sigma)}{s}}_{(\text{II})}$$

where (I) is the oracle gap (irreducible), and (II) is LVSD-induced (vanishes as $s \to \infty$). We define *Local-View Semantic Drift (LVSD)* as the finite-$s$ risk of class inversion due

to crop-level label variance, formally characterized by the event $\mathcal{E}_{s,c} := \{\hat{p}_s(c) \geq \hat{p}_s(y)\}$ for $c \neq y$. When $\bar{p}_y > \bar{p}_c$, letting $v_{s,c} := (\Sigma_{yy} + \Sigma_{cc} - 2\Sigma_{yc})/s$, the Cantelli inequality yields the distribution-free bound

$$\Pr(\mathcal{E}_{s,c}) \leq \frac{v_{s,c}}{v_{s,c} + (\bar{p}_y - \bar{p}_c)^2},$$

which is monotonically decreasing in $s$ and vanishes as $s \to \infty$. LVSD is present whenever $\Sigma \neq 0$ and $s < \infty$, and is absent iff $\Pr(\mathcal{E}_{s,c}) = 0$ for all $c \neq y$.

**Lemma 3.2.** *For $s$ i.i.d. crops define $\hat{p}_s := \frac{1}{s} \sum_{i=1}^{s} \tilde{p}(x_i^{(\text{crop})})$. Then,*

$$\mathbb{E}[\hat{p}_s] = \bar{p}, \quad \text{Cov}(\hat{p}_s) = \frac{\Sigma}{s}, \quad \mathbb{E}[\|\hat{p}_s - \bar{p}\|_2^2] = \frac{\text{Tr}(\Sigma)}{s}.$$

*In particular, under LVSD, the deviation is strictly positive for any finite $s$ and decays as $\mathcal{O}(1/s)$.*

**Definition 3.3** (Soft Label per Image (SLI)). The soft labels per image (SLI) denote the number of augmented soft labels (e.g., crops or views) generated for each image.

**Definition 3.4** (Soft Label per Class (SLC)). Let $C \in \mathbb{N}$ be the number of classes, and ipc the number of images per class. Given that each image has SLI soft labels, the total number of soft labels per class (SLC) is defined as

$$\text{SLC} = \text{ipc} \times \text{SLI}.$$

Each soft label is a $C$-dimensional vector, with each scalar entry stored using $b$ bits. The corresponding per-class storage budget (in bits) is therefore

$$\text{Storage}(\text{SLC}) = \text{SLC} \cdot (Cb).$$

***Discussion (Why SLC).*** SLC quantifies per-class pre-generated supervision. Fixing SLC controls label-side storage: regardless of IPC, equal SLC yields the same number of stored soft labels per class. By contrast, pruning ratios alone are confounded by IPC and obscure absolute storage.

To reduce the storage overhead of soft-label supervision, LPLD (Xiao & He, 2024) limits the total number of stored teacher predictions and reuses them during training. We refer to this storage budget as SLC (Definition 3.4). While this substantially reduces storage, Lemma 3.2 implies that finite-$s$ supervision deviates from the full-coverage regime due to *Local-View Semantic Drift* (Definition 3.1). In what follows, we quantify this deviation.

**Deviation from the Ideal Optimization Objective.** By Theorem 3.5, *Local-View Semantic Drift*, i.e., nonzero per-crop prediction covariance, induces a *strictly positive* lower bound on the expected mismatch between $\mathcal{L}_s$ and $\mathcal{L}_{\text{ideal}}$ of order $\Theta(s^{-1/2})$, with a distribution-dependent constant $C(\sigma, \kappa)$. Consequently, in low-SLC regimes, the finite-SLC objective is systematically misaligned with the ideal supervision goal, the gap vanishes only as $s \to \infty$.

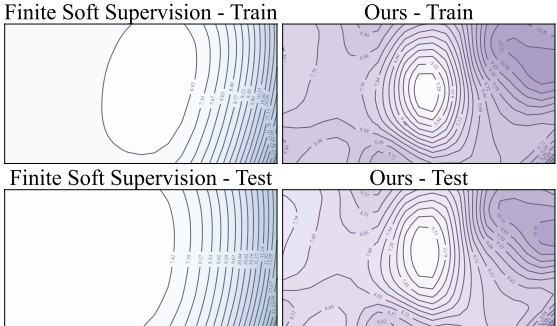

| Finite Soft Supervision - Train | Ours - Train |
| Finite Soft Supervision - Test | Ours - Test |

*Figure 2.* Train and test loss landscapes on an IPC=10 distilled dataset with SLC=50, comparing (i) finite soft-label coverage and (ii) our method.

**Theorem 3.5** (Proof in Appendix B.1). *Consider a synthetic image $\tilde{x}$ with an augmentation distribution $\mathcal{T}(\tilde{x})$. Each crop $\tilde{x}_i^{(\text{crop})} \sim \mathcal{T}(\tilde{x})$ is assigned a teacher soft label $\tilde{p}_i \in \Delta^C$, while the student model produces a predictive distribution $q_\theta(\cdot \mid \tilde{x}_i^{(\text{crop})})$. Let $\mathcal{L}[\tilde{p}, q] : \Delta^C \times \Delta^C \to \mathbb{R}_{\geq 0}$ denote a per-crop loss functional. The empirical loss over $s$ independent crops is defined as $\mathcal{L}_s(\theta; \tilde{x}) := \frac{1}{s} \sum_{i=1}^{s} \mathcal{L}\left[\tilde{p}_i, q_\theta(\cdot \mid \tilde{x}_i^{(\text{crop})})\right]$, and the ideal loss under full augmentation coverage is defined as $\mathcal{L}_{\text{ideal}}(\theta; \tilde{x}) := \mathbb{E}_{\tilde{x}^{(\text{crop})} \sim \mathcal{T}(\tilde{x})}\left[\mathcal{L}\left[\tilde{p}, q_\theta(\cdot \mid \tilde{x}^{(\text{crop})})\right]\right]$. For notational simplicity, we omit the explicit dependence on $(\theta, \tilde{x})$ when the context is clear. Define the variance and the normalized fourth central moment of the per-crop loss as*

$$\sigma^2 := \text{Var}_{\tilde{x}^{(\text{crop})} \sim \mathcal{T}(\tilde{x})}\left[\mathcal{L}\left[\tilde{p}, q_\theta(\cdot \mid \tilde{x}^{(\text{crop})})\right]\right] < \infty,$$
$$\kappa := \frac{\mathbb{E}\left[\left(\mathcal{L} - \mathbb{E}\mathcal{L}\right)^4\right]}{\sigma^4} \in [1, \infty). \tag{2}$$

*Assume that $\sigma^2 < \infty$ and $\kappa < \infty$. Then the expected deviation between the empirical and ideal losses satisfies*

$$\mathbb{E}\left[\left|\mathcal{L}_s - \mathcal{L}_{\text{ideal}}\right|\right] \geq \frac{\sigma}{\sqrt{s}} \cdot \frac{16}{25\sqrt{5}} \cdot \min\left\{\frac{1}{\kappa}, \frac{1}{3}\right\}. \tag{3}$$

**Few Soft Labels Make Train-Test Misaligned.** Let $\hat{\theta}_\star := \arg\min_\theta \mathcal{L}_{\text{ideal}}(\theta)$ denote the *oracle* obtained under exhaustive local-view supervision from a strong teacher. By construction, $\hat{\theta}_\star$ *maximally aligns* with the teacher's predictive distribution across local views, we assume it achieves the best attainable generalization. Thus any deviation $\hat{\theta}_s \neq \hat{\theta}_\star$ may degrade generalization. We therefore study the excess loss $\mathbb{E}\left[\mathcal{L}_{\text{ideal}}(\hat{\theta}_s) - \mathcal{L}_{\text{ideal}}(\hat{\theta}_\star)\right]$, which is nonnegative by the optimality of $\hat{\theta}_\star$ for $\mathcal{L}_{\text{ideal}}$ and vanishes iff $\hat{\theta}_s = \hat{\theta}_\star$. Under limited soft-label coverage (small $s$), $\mathcal{L}_s$ exhibits *LVSD* and optimizes a proxy of $\mathcal{L}_{\text{ideal}}$; consequently $\hat{\theta}_s$ departs from $\hat{\theta}_\star$, incurring an unavoidable generalization penalty. Theorem 3.6 formalizes this effect, yielding a lower bound of order $\Omega(1/s)$ that disappears only as $s \to \infty$.

**Theorem 3.6** (Proof in Appendix B.2). *Let $\mathcal{L}_{\text{ideal}}(\theta)$ be twice continuously differentiable in a neighborhood $\mathcal{N}$ of its unique minimizer $\hat{\theta}_\star$, and denote $H_\star := \nabla^2 \mathcal{L}_{\text{ideal}}(\hat{\theta}_\star) \succeq \mu I$ for some $\mu > 0$. Write $g(\theta; x) := \nabla_\theta \ell(\theta; x)$ so that $\nabla \mathcal{L}_{\text{ideal}}(\theta) = \mathbb{E}[g(\theta; x)]$, and let $\hat{\theta}_s \in \arg\min_\theta \mathcal{L}_s(\theta)$ be any ERM. Assume: (A1) (Unbiased score & covariance) $\mathbb{E}[g(\hat{\theta}_\star; x)] = 0$, and $\Sigma_\star := \text{Cov}(g(\hat{\theta}_\star; x))$ with $\mathbb{E}\|g(\hat{\theta}_\star; x)\|^{2+\kappa} < \infty$ for some $\kappa > 0$. (A2) (Hessian Lipschitz) $\nabla^2 \mathcal{L}_{\text{ideal}}$ is $L_H$-Lipschitz on $\mathcal{N}$. (A3) (Local uniform concentration) There exist $r_0 > 0$ and constants $C_{\text{uc}} > 0$, $\bar{c} > 0$ such that for all $s$ and $\delta \in (0, 1)$, with probability at least $1 - \delta$, $\sup_{\theta \in \mathbb{B}(\hat{\theta}_\star, r_0)} \left\|H_s(\theta) - \nabla^2 \mathcal{L}_{\text{ideal}}(\theta)\right\| \leq C_{\text{uc}} \sqrt{\frac{\log(1/\delta)}{s}}$, $H_s(\theta) := \frac{1}{s} \sum_{i=1}^{s} \nabla_\theta^2 \ell(\theta; x_i)$, and $\sup_{\theta \in \mathbb{B}(\hat{\theta}_\star, r_0)} \left\|(\nabla \mathcal{L}_s - \nabla \mathcal{L}_{\text{ideal}})(\theta) - (\nabla \mathcal{L}_s - \nabla \mathcal{L}_{\text{ideal}})(\hat{\theta}_\star)\right\| \leq \bar{c} \sqrt{\frac{\log(1/\delta)}{s}} \|\theta - \hat{\theta}_\star\|$. (A4) (ERM stays local) There exists a sequence $\delta_s \downarrow 0$ such that $\Pr\left(\hat{\theta}_s \in \mathbb{B}(\hat{\theta}_\star, r_0)\right) \geq 1 - \delta_s$. (A5) (Boundedness near optimum) There exists $B < \infty$ such that $\mathcal{L}_{\text{ideal}}(\theta) \leq B$ for all $\theta \in \mathbb{B}(\hat{\theta}_\star, r_0)$. See more details about assumptions in Appendix B.2. Then there exist constants $C_1, C_2, C_b > 0$ depending only on $(\mu, L_H)$, such that for all $s$,*

$$\mathbb{E}\left[\mathcal{L}_{\text{ideal}}(\hat{\theta}_s) - \mathcal{L}_{\text{ideal}}(\hat{\theta}_\star)\right] \geq \frac{1}{2s} \text{tr}\left(H_\star^{-1} \Sigma_\star\right) - \frac{C_1}{s^{3/2}} - \frac{C_2}{s^2} - C_b \, \delta_s. \tag{4}$$

**Visualization of the Limitations of Limited Soft Label Supervision.** To illustrate the limitations of limited soft-label coverage, we compare the model's behavior on both the training and test sets, as shown in Fig. 2. Under finite soft-label supervision, the test-time loss landscape deviates notably from that of the training set, indicating overfitting and reduced generalization.

### 3.2. Calibrating LVSD with Accurate Supervision

To mitigate *LVSD* under finite-$s$ soft-label coverage, we propose **H**ard **L**abel to **A**lleviate **L**ocal **S**emantic **D**rift (**HALD**), a *soft→hard→soft* calibration schedule. The student first learns coarse discriminative features from finite-$s$ soft labels, improving soft–hard gradient alignment. Hard-label supervision then enforces class-consistent constraints to suppress crop-induced variance. Finally, teacher-guided training resumes on the variance-reduced representation, balancing limited soft-label guidance with global semantic supervision and improving overall performance.

**How to determine the training duration for each stage.** We assume (as in our theoretical framework) that the model can fit the finite-$s$ soft-label supervision on $\Omega_{\text{soft}}$ to empirical risk minimization (ERM). Let $n_{\text{soft}}$ denote the epoch budget required for the model to reach convergence on $\Omega_{\text{soft}}$, and let $n_{\text{total}}$ be the total training budget. We allocate the remaining

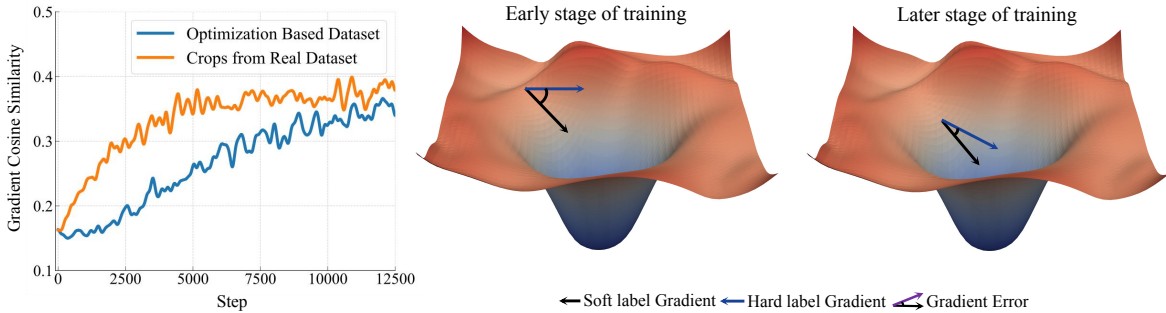

*Figure 3.* Gradient similarity between hard- and soft-label losses over training, evaluated on real-image crops and optimization-based distilled data, showing a clear upward trend indicative of strengthened alignment.

epochs to hard-label calibration: $n_{\text{hard}} := n_{\text{total}} - n_{\text{soft}}$ ($\geq 0$). Training then follows:

$$\underbrace{T_A = \left\lfloor \frac{n_{\text{soft}}}{2} \right\rfloor}_{\text{soft}}, \quad \underbrace{T_B = n_{\text{hard}}}_{\text{hard}}, \quad \underbrace{T_C = n_{\text{soft}} - T_A}_{\text{soft}},$$

If $n_{\text{total}} \leq n_{\text{soft}}$, we set $n_{\text{hard}} = 0$ and run soft-label training only. This schedule preserves the ERM fit on $\Omega_{\text{soft}}$, inserts a hard-label calibration phase of length $n_{\text{hard}}$ to mitigate local semantic drift, and finally re-aligns with $\Omega_{\text{soft}}$ to consolidate the variance reduction benefits.

*(i) Stage A (soft pretraining).* Let $s$ denote the *total* number of pre-generated soft labels (SLC × C). Define the global soft-label pool:

$$\Omega_{\text{soft}} := \left\{ (x_i^{(\text{crop})}, \tilde{p}_i) \right\}_{i=1}^{s}, \qquad \tilde{p}_i := \tilde{p}(\cdot \mid x_i^{(\text{crop})}),$$

where each $x_i^{(\text{crop})}$ is obtained by sampling a training image $\tilde{x}$ and a crop $x \sim \mathcal{T}(\tilde{x})$. At step $t$, sample indices $J_t = \{j_1, \ldots, j_B\} \subset \{1, \ldots, s\}$ *with replacement* and form the mini-batch $\{(x_{j_b}^{(\text{crop})}, \tilde{p}_{j_b})\}_{b=1}^{B}$. Using the same per-crop loss $\mathcal{L}[\cdot, \cdot]$, minimize the batch estimator,

$$\widehat{\mathcal{L}}_{\text{soft}}^{(t)}(\theta) = \frac{1}{B} \sum_{b=1}^{B} \mathcal{L}\big(\tilde{p}_{j_b}, q_\theta(\cdot \mid x_{j_b}^{(\text{crop})})\big),$$

$$\theta_{t+1} = \theta_t - \eta_t \nabla_\theta \widehat{\mathcal{L}}_{\text{soft}}^{(t)}(\theta_t).$$

We denote by $\hat{\theta}_s^{\text{A}}$ the parameters obtained after Stage A training using this pool-sampling procedure.

*(ii) Stage B (de-LVSD via hard labels).* We use *hard labels* to refer to targets globally anchored to the ground-truth class $y$ regardless of local crop content, in contrast to crop-dependent teacher soft labels. Since the synthetic data has less clear semantics than real data, stronger label flattening is needed for stable calibration; we thus employ label-smoothed targets, which we term *heavily-flattened hard labels*, rather than strict one-hot vectors. Define label smoothing and the CutMix target (for $C$ classes) as: $\text{LS}_\alpha(y) = (1 -$

$\alpha) \delta_y + \alpha \frac{1}{C}$, and $t_{\lambda,\alpha}(y, y') = (1-\lambda) \text{LS}_\alpha(y) + \lambda \text{LS}_\alpha(y')$. Let the sampling space be:

$$\Omega_{\text{cal}} := \left\{ ((\tilde{x}, y), (\tilde{x}', y'), x, x', \lambda, m) : \begin{array}{l} x \sim \mathcal{T}(\tilde{x}), \\ x' \sim \mathcal{T}(\tilde{x}'), \\ \lambda \in (0, 1), \\ m \in \mathcal{M} \end{array} \right\}.$$

For any $\omega = ((\tilde{x}, y), (\tilde{x}', y'), x, x', \lambda, m) \in \Omega_{\text{cal}}$, define the calibration loss,

$$\ell_{\text{cal}}(\theta; \omega) := \mathcal{L}\Big(t_{\lambda,\alpha}(y, y'), q_\theta(\cdot \mid \text{CM}_{\lambda,m}(x, x'))\Big).$$

Initialize $\theta_0 := \hat{\theta}_s^{\text{A}}$. At each step $t$, draw an i.i.d. minibatch $\{\omega_i^{(t)}\}_{i=1}^{B} \subset \Omega_{\text{cal}}$ and update,

$$\widehat{\mathcal{L}}_{\text{cal}}^{(t)}(\theta) = \frac{1}{B} \sum_{i=1}^{B} \ell_{\text{cal}}(\theta; \omega_i^{(t)}), \quad \theta_{t+1} = \theta_t - \eta_t \nabla_\theta \widehat{\mathcal{L}}_{\text{cal}}^{(t)}(\theta_t).$$

As crops and CutMix geometry are resampled at every step, minibatches are effectively non-repeating and provide ground-truth–anchored, diverse local views of each base image, thereby mitigating the semantic bias induced by finite-$s$ soft-label supervision in Stage A.

*(iii) Stage C (soft refinement).* Initialize from $\hat{\theta}^{\text{B}}$; Stage C *follows Stage A's* pool-based protocol (same sampler on $\Omega_{\text{soft}}$ and per-crop loss $\mathcal{L}$), yielding final $\hat{\theta}$.

### 3.3. Theoretical Analysis for HALD

**Optimization Coherence and Stability.** Theorem 3.7 shows that soft–hard gradient alignment is governed by the ratio $D/m_0$, where $D$ is the inter-class gradient spread and $m_0$ the minimal gradient norm. As training converges, representations stabilize and classifier alignment improves, causing $D$ to decay faster than $m_0$ and tightening the alignment bound. Consistently, Fig. 3 shows a positive and steadily increasing cosine similarity between soft- and hard-label gradients, validating the theory. The *Soft–Hard–Soft* design naturally follows from this theory: the first Soft stage

*Table 1.* Comparison with SOTA methods on Tiny-ImageNet.

| | SLI = 2 | | | | | SLI = 1 | | | | |
| --- | --- | --- | --- | --- | --- | --- | --- | --- | --- | --- |
| | SRe$^2$L | RDED | FADRM | LPLD | Ours | SRe$^2$L | RDED | FADRM | LPLD | Ours |
| IPC=10 | 14.6±0.2 | 12.5±0.3 | 17.4±0.5 | 13.3±0.3 | **22.8**±0.3 | 8.3±0.4 | 7.7±0.2 | 10.1±0.3 | 8.2±0.2 | **18.6**±0.4 |
| *Storage* | SLC = 20 (1.52 MB) | | | | | SLC = 10 (0.76 MB) | | | | |
| IPC=20 | 21.8±0.3 | 19.6±0.4 | 26.4±0.2 | 21.3±0.4 | **29.7**±0.5 | 14.8±0.3 | 11.7±0.2 | 17.5±0.2 | 14.0±0.3 | **25.9**±0.3 |
| *Storage* | SLC = 40 (3.04 MB) | | | | | SLC = 20 (1.52 MB) | | | | |
| IPC=30 | 27.3±0.5 | 23.6±0.6 | 31.0±0.4 | 27.5±0.5 | **33.8**±0.4 | 19.7±0.2 | 17.9±0.5 | 23.8±0.4 | 17.4±0.4 | **28.7**±0.3 |
| *Storage* | SLC = 60 (4.56 MB) | | | | | SLC = 30 (2.28 MB) | | | | |
| IPC=40 | 29.6±0.2 | 26.8±0.5 | 32.9±0.4 | 29.1±0.4 | **35.2**±0.3 | 22.2±0.3 | 19.6±0.3 | 26.6±0.3 | 22.1±0.5 | **30.3**±0.4 |
| *Storage* | SLC = 80 (6.08 MB) | | | | | SLC = 40 (3.04 MB) | | | | |
| IPC=50 | 31.9±0.2 | 27.9±0.4 | 36.0±0.4 | 34.3±0.3 | **38.2**±0.5 | 24.0±0.4 | 20.5±0.2 | 27.8±0.5 | 24.1±0.2 | **30.7**±0.4 |
| *Storage* | SLC = 100 (7.60 MB) | | | | | SLC = 50 (3.80 MB) | | | | |

*Table 2.* Comparison with SOTA methods on ImageNet-1K. [†] denotes the reported results.

| | SLI = 10 | | | | | SLI = 5 | | | | |
| --- | --- | --- | --- | --- | --- | --- | --- | --- | --- | --- |
| | SRe$^2$L | RDED | FADRM | LPLD | Ours | SRe$^2$L | RDED | FADRM | LPLD | Ours |
| IPC=10 | 25.9±0.2 | 19.9±0.5 | 27.9±0.3 | 23.1[†]±0.1 | **35.6**±0.3 | 14.5±0.2 | 11.8±0.6 | 16.1±0.3 | 14.5±0.3 | **30.3**±0.4 |
| *Storage* | SLC = 100 (190 MB) | | | | | SLC = 50 (95 MB) | | | | |
| IPC=20 | 35.1±0.3 | 29.1±0.4 | 40.1±0.3 | 35.9[†]±0.3 | **46.5**±0.4 | 25.4±0.4 | 21.0±0.4 | 29.5±0.5 | 24.0±0.4 | **42.9**±0.3 |
| *Storage* | SLC = 200 (380 MB) | | | | | SLC = 100 (190 MB) | | | | |
| IPC=30 | 40.1±0.4 | 37.2±0.5 | 46.6±0.6 | 42.0±0.2 | **50.7**±0.2 | 30.9±0.4 | 25.9±0.3 | 37.1±0.3 | 31.9±0.2 | **46.5**±0.4 |
| *Storage* | SLC = 300 (570 MB) | | | | | SLC = 150 (285 MB) | | | | |
| IPC=40 | 43.3±0.2 | 40.9±0.6 | 50.0±0.5 | 44.9±0.4 | **52.6**±0.3 | 35.1±0.2 | 32.1±0.5 | 41.4±0.6 | 36.3±0.3 | **48.7**±0.3 |
| *Storage* | SLC = 400 (760 MB) | | | | | SLC = 200 (380 MB) | | | | |
| IPC=50 | 46.8±0.3 | 43.5±0.5 | 52.7±0.6 | 48.6[†]±0.2 | **53.7**±0.2 | 39.5±0.2 | 34.9±0.5 | 45.5±0.4 | 39.4±0.3 | **49.5**±0.2 |
| *Storage* | SLC = 500 (950 MB) | | | | | SLC = 250 (475 MB) | | | | |

aligns the model and strengthens gradient similarity; the **Hard stage** reduces variance and corrects semantic drift; and the final stage restores fine-grained teacher consistency on the variance-reduced representation.

**Theorem 3.7** (Soft–Hard Gradient Consistency; proof in Appendix B.3.1). *Fix a crop $x^{(\text{crop})} \sim \mathcal{T}(\tilde{x})$ and C classes. Let $g_c := \nabla_\theta \log q_\theta(c \mid x^{(\text{crop})})$, $\nabla_\theta \mathcal{L}_{\text{soft}} = -\sum_c \tilde{p}_c g_c$, $\nabla_\theta \mathcal{L}_{\text{hard}} = -\sum_c \bar{p}_c^{(\alpha)} g_c$ (where $\bar{p}^{(\alpha)}$ is the $\alpha$-smoothed one–hot). Assume $D := \sup_{i \neq j} \|g_i - g_j\| < \infty$ and $m_0 := \min\{\|\sum_c \tilde{p}_c g_c\|, \|\sum_c \bar{p}_c^{(\alpha)} g_c\|\} > 0$. Then there exists a constant $C_{\text{align}}(\tilde{x}, \alpha)$ depending only on the teacher's predictive entropy and the smoothing rate such that,*

$$\mathbb{E}_{\text{crop}}[\cos(\nabla_\theta \mathcal{L}_{\text{soft}}, \nabla_\theta \mathcal{L}_{\text{hard}})] \geq 1 - \frac{D}{m_0} \cdot C_{\text{align}}(\tilde{x}, \alpha).$$

**Corollary 3.8** (Proof in Appendix B.4). *Assume the conditions of Theorem 3.7 hold so that $\mathbb{E}[\langle u, v \rangle] = \mathbb{E}[\cos(v_{\text{soft}}, v_{\text{hard}})] \geq \rho_\star$, where $u := v_{\text{soft}}/\|v_{\text{soft}}\|$ and $v := v_{\text{hard}}/\|v_{\text{hard}}\|$. Let $(u_i, v_i)_{i=1}^s$ be i.i.d. copies of $(u, v)$. Then the effective sample size satisfies:*

$$s_{\text{eff}} \geq \frac{s}{1 - \rho_\star^2}. \tag{5}$$

**Analysis of Hard Label Calibration.** By Corollary 3.8, hard-label calibration increases the effective sample size from $s$ to at least $s_{\text{eff}}$, thereby improving the optimization objective in Equations 3 and the generalization bound in

Equation 4 via variance reduction. Performance gain is visualized in Fig. 2. *Intuitively*, hard-label calibration mitigates local semantic drift by enlarging the effective sample size, which reduces the sample variance and alleviates overfitting from finite-$s$ soft-label supervision. This improvement is driven by the strong alignment between soft- and hard-label gradients (high expected cosine similarity), ensuring that optimization on hard labels remains informative about unseen crops drawn from the same population distribution.

## 4. Experiment

### 4.1. Experiment Settings

**Datasets.** We evaluate **HALD** on Tiny-ImageNet ($64 \times 64$, $C=200$) (Le & Yang, 2015) and ImageNet-1K ($224 \times 224$, $C=1000$) (Deng et al., 2009), two widely-adopted benchmarks for dataset distillation that together span both low- and high-resolution regimes with markedly different class scales, enabling a comprehensive assessment of the scalability and generalization of our method across diverse settings.

**Generation Methods.** We consider four representative paradigms that collectively cover the synthetic–real and low–high diversity axes: (i) SRe$^2$L (Yin et al., 2023), an optimization-based approach that synthesizes images from a pretrained model but suffers from limited intra-class diversity; (ii) LPLD (Xiao & He, 2024), which explicitly improves the diversity of SRe$^2$L; (iii) RDED (Sun et al.,

2024), a selection-based method that constructs distilled data from class-preserving crops of real images; and (iv) FADRM (Cui et al., 2025a), a residual-hybrid approach that fuses real-image priors with optimized synthetic images. Spanning the spectrum from fully synthetic to real-data selection and from low to high diversity, these methods provide a comprehensive basis for comparison.

**Baseline Methods.** We compare the proposed training paradigm (**HALD**) with three baselines: (i) *Soft-Only*, which uses only soft-label supervision; (ii) GIFT (Shang et al., 2025a), which directly integrates hard-label information into soft labels; and (iii) Joint Objective, which optimizes a combined loss $\mathcal{L} = \mathcal{L}_{\text{soft}} + \lambda \mathcal{L}_{\text{hard}}$, where $\lambda$ balances the two supervision sources. Unless otherwise specified, all generation methods use the strongest baseline (*Soft-Only*) as training, while our approach adopts FADRM for generation and **HALD** for training.

## 4.2. Main Result

As shown in Table 1 and Table 2, our method consistently achieves superior performance. With SLI= 5 and a 50-IPC distilled dataset (SLC=250), **HALD** reaches 49.5% Top-1 accuracy on ImageNet-1K, surpassing the previous SOTA LPLD by +10.1%, thereby validating its effectiveness.

*Table 3.* Performance comparison under identical storage and training budgets, highlighting HALD's advantage through stage-wise soft–hard integration. JO. denotes the Joint Objective method.

| SLC | GIFT | JO. $\lambda = 1$ | JO. $\lambda = 0.1$ | JO. $\lambda = 0.01$ | Soft Only | Ours |
|---|---|---|---|---|---|---|
| 100 | 27.0 | 5.9 | 7.0 | 13.1 | 26.9 | **43.5** |
| 200 | 39.1 | 8.1 | 9.4 | 17.5 | 39.2 | **47.3** |
| 300 | 46.7 | 9.5 | 10.3 | 20.1 | 46.6 | **50.7** |

**Comparison with more baselines.** As shown in Table 3, **HALD** achieves the best overall performance, while GIFT is comparable to the soft-only baseline. For the Joint Objective, performance peaks at $\lambda = 0$ and degrades with increasing $\lambda$, indicating gradient inconsistency from mixing hard and soft supervision. By contrast, HALD's stage-wise design mitigates this conflict and yields consistent gains. We further compare **HALD** with LPQLD (Xiao & He, 2026) in Table 4, where the two rows correspond to IPC=10 and IPC=20 respectively, and **HALD** consistently outperforms LPQLD across both settings.

*Table 4.* Comparison with LPQLD.

| LPQLD | HALD |
|---|---|
| 32.8 | **36.2** |
| 42.3 | **43.9** |

**Storage Efficiency.** Table 5 reports performance under varying soft label storage budgets. While LPLD degrades sharply with tighter constraints, **HALD** maintains strong accuracy (e.g., 36.9% at 95M, a +22.6% improvement over LPLD at the same budget). This demonstrates the storage efficiency of our calibration strategy, which effectively enhances the utility of stored soft labels under limited capacity.

**Results on more datasets.** To evaluate generalization beyond Tiny-ImageNet and ImageNet-1K, we additionally test **HALD** on CIFAR-100 and ImageWoof, covering both general and fine-grained recognition settings. As shown in Table 6, **HALD** consistently outperforms the Soft-Only baseline across all settings.

*Table 5.* Storage *vs.* Effectiveness. (ImageNet-1K IPC=50)

|  | 570M | 475M | 380M | 285M | 190M | 95M |
|---|---|---|---|---|---|---|
| LPLD | 43.1 | 39.4 | 36.9 | 33.7 | 25.5 | 14.3 |
| **Ours** | 50.3 [↑7.2] | 49.5 [↑10.1] | 47.7 [↑10.8] | 42.7 [↑9.0] | 40.9 [↑15.4] | 36.9 [↑22.6] |

*Table 6.* Results on CIFAR-100 and ImageWoof under IPC=10.

| Dataset | SLC | Soft-Only | HALD |
|---|---|---|---|
| CIFAR-100 | 10 | 10.6 | **21.0** [↑10.4] |
|  | 20 | 16.5 | **26.0** [↑9.5] |
| ImageWoof | 10 | 23.5 | **27.4** [↑3.9] |
|  | 20 | 27.4 | **29.3** [↑1.9] |

## 4.3. Ablation

**How Long to Use Hard Labels.** We validate our theoretical assumption that the total soft-label phase should match the convergence time of standalone soft-label training. As shown in Table 7, extending this phase to its full predefined length consistently improves results.

*Table 7.* Impact of soft-label phase length on performance.

| Method | Soft-Label Phase Length (epochs) | | | |
|---|---|---|---|---|
|  | 100 | 150 | **200** | 250 |
| FADRM | 31.3 | 34.8 | **35.6** | 29.3 |
| RDED | 16.6 | 23.5 | **24.4** | 22.5 |
| LPLD | 24.1 | 27.1 | **30.0** | 26.3 |
| SRe$^2$L | 26.7 | 30.9 | **31.9** | 26.4 |

*Soft-label convergence length = **200** epochs*

**Impact of Hard-Label Calibration.** To assess the generality of HALD, we compare the *Soft–Hard–Soft* (ours) schedule with *Soft–Only* across multiple distillation techniques (Table 8). Consistent gains across methods confirm the benefit of hard-label calibration, especially under low-SLC regimes where LVSD is more pronounced.

**When to Switch to Hard Labels.** To assess the effectiveness of the proposed paradigm, we evaluate **HALD** under four training schedules. As shown in Table 9, *Soft–Hard–Soft* achieves the highest accuracy, indicating that introducing hard labels mid-training is most effective, consistent with Theorem 3.7 and its prediction of stronger gradient alignment after partial training.

**Effect of the first and last soft-label stages.** As shown in Table 10, a balanced allocation between the first and last soft-label phases yields the best performance. Overweighting the first phase leaves insufficient budget for the post-calibration recovery of fine-grained teacher semantics, whereas overweighting the last phase initiates hard-label calibration be-

*Table 8.* Comprehensive ablation of the impact of incorporating hard-label supervision across state-of-the-art dataset distillation methods on ImageNet-1K and Tiny-ImageNet. All models are trained for 300 epochs under identical hyperparameters, with the evaluation protocol being the sole difference. $^{\dagger}$ denotes values reported by the corresponding original sources.

| IPC | Generation | Evaluation | ImageNet-1K | | | | | | Tiny-ImageNet | |
| --- | --- | --- | --- | --- | --- | --- | --- | --- | --- | --- |
| | | | SLC=300 | SLC=250 | SLC=200 | SLC=150 | SLC=100 | SLC=50 | SLC=100 | SLC=50 |
| IPC=10 | SRe²L | Soft-Only | 37.1 | 36.5 | 34.8 | 30.4 | 25.9 | 14.5 | 31.4 | 24.0 |
| | | Ours | 37.5 $^{\uparrow 0.4}$ | 37.3 $^{\uparrow 0.8}$ | 37.0 $^{\uparrow 2.2}$ | 33.8 $^{\uparrow 3.4}$ | 31.9 $^{\uparrow 6.0}$ | 26.7 $^{\uparrow 12.2}$ | 31.9 $^{\uparrow 0.5}$ | 25.5 $^{\uparrow 1.5}$ |
| | RDED | Soft-Only | 29.8 | 28.5 | 28.3 | 25.7 | 19.9 | 11.8 | 27.6 | 22.3 |
| | | Ours | 30.4 $^{\uparrow 0.6}$ | 29.2 $^{\uparrow 0.7}$ | 29.8 $^{\uparrow 1.5}$ | 25.9 $^{\uparrow 0.2}$ | 24.4 $^{\uparrow 4.5}$ | 20.9 $^{\uparrow 9.1}$ | 31.0 $^{\uparrow 3.4}$ | 27.0 $^{\uparrow 4.7}$ |
| | LPLD | Soft-Only | 32.7 $^{\dagger}$ | 35.1 | 32.3 | 28.6$^{\dagger}$ | 23.1$^{\dagger}$ | 14.5 | 31.5 | 23.5 |
| | | Ours | 36.2 $^{\uparrow 3.5}$ | 36.7 $^{\uparrow 1.6}$ | 34.4 $^{\uparrow 2.1}$ | 30.8 $^{\uparrow 2.2}$ | 30.0 $^{\uparrow 6.9}$ | 26.3 $^{\uparrow 11.8}$ | 32.6 $^{\uparrow 1.1}$ | 26.5 $^{\uparrow 3.0}$ |
| | FADRM | Soft-Only | 42.0 | 41.4 | 39.0 | 34.1 | 27.9 | 16.1 | 34.4 | 28.1 |
| | | Ours | 43.0 $^{\uparrow 1.0}$ | 42.0 $^{\uparrow 0.6}$ | 40.7 $^{\uparrow 1.7}$ | 39.0 $^{\uparrow 4.9}$ | 35.6 $^{\uparrow 7.7}$ | 30.3 $^{\uparrow 14.2}$ | 36.2 $^{\uparrow 1.8}$ | 30.7 $^{\uparrow 2.6}$ |
| IPC=20 | SRe²L | Soft-Only | 39.2 | 38.1 | 35.1 | 29.4 | 25.4 | 12.8 | 30.9 | 22.0 |
| | | Ours | 42.3 $^{\uparrow 3.1}$ | 41.9 $^{\uparrow 3.8}$ | 40.7 $^{\uparrow 5.6}$ | 37.3 $^{\uparrow 7.9}$ | 35.9 $^{\uparrow 10.5}$ | 27.5 $^{\uparrow 14.7}$ | 32.7 $^{\uparrow 1.8}$ | 24.5 $^{\uparrow 2.5}$ |
| | RDED | Soft-Only | 35.2 | 33.2 | 29.1 | 26.6 | 21.0 | 10.8 | 30.1 | 20.7 |
| | | Ours | 39.8 $^{\uparrow 4.6}$ | 38.3 $^{\uparrow 5.1}$ | 36.8 $^{\uparrow 7.7}$ | 34.6 $^{\uparrow 8.0}$ | 33.5 $^{\uparrow 12.5}$ | 24.1 $^{\uparrow 13.3}$ | 33.0 $^{\uparrow 2.9}$ | 25.2 $^{\uparrow 4.5}$ |
| | LPLD | Soft-Only | 41.0$^{\dagger}$ | 38.5 | 35.9$^{\dagger}$ | 33.0$^{\dagger}$ | 24.0 | 11.7 | 35.2 | 21.2 |
| | | Ours | 43.9 $^{\uparrow 2.9}$ | 40.8 $^{\uparrow 2.3}$ | 40.3 $^{\uparrow 4.4}$ | 38.3 $^{\uparrow 5.3}$ | 35.9 $^{\uparrow 11.9}$ | 24.9 $^{\uparrow 13.2}$ | 36.0 $^{\uparrow 0.8}$ | 24.6 $^{\uparrow 3.4}$ |
| | FADRM | Soft-Only | 45.9 | 44.3 | 40.1 | 35.3 | 29.5 | 13.9 | 37.0 | 26.8 |
| | | Ours | 48.3 $^{\uparrow 2.4}$ | 47.7 $^{\uparrow 3.4}$ | 46.5 $^{\uparrow 6.4}$ | 44.0 $^{\uparrow 8.7}$ | 42.9 $^{\uparrow 13.4}$ | 32.7 $^{\uparrow 18.8}$ | 37.6 $^{\uparrow 0.6}$ | 29.8 $^{\uparrow 3.0}$ |
| IPC=50 | SRe²L | Soft-Only | 41.2 | 39.5 | 36.3 | 30.6 | 25.2 | 14.4 | 31.9 | 24.0 |
| | | Ours | 43.5 $^{\uparrow 2.3}$ | 42.3 $^{\uparrow 2.8}$ | 41.5 $^{\uparrow 5.2}$ | 35.2 $^{\uparrow 4.6}$ | 32.7 $^{\uparrow 7.5}$ | 29.7 $^{\uparrow 15.3}$ | 32.9 $^{\uparrow 1.0}$ | 26.0 $^{\uparrow 2.0}$ |
| | RDED | Soft-Only | 37.3 | 34.9 | 31.9 | 27.3 | 21.0 | 12.6 | 27.9 | 20.5 |
| | | Ours | 46.0 $^{\uparrow 8.7}$ | 45.1 $^{\uparrow 10.2}$ | 43.4 $^{\uparrow 11.5}$ | 42.3 $^{\uparrow 15.0}$ | 38.9 $^{\uparrow 17.9}$ | 35.5 $^{\uparrow 22.9}$ | 31.1 $^{\uparrow 3.2}$ | 26.4 $^{\uparrow 5.9}$ |
| | LPLD | Soft-Only | 43.1$^{\dagger}$ | 39.4 | 36.9 | 33.7$^{\dagger}$ | 25.5 | 14.3 | 34.4 | 24.1 |
| | | Ours | 44.9 $^{\uparrow 1.8}$ | 43.6 $^{\uparrow 4.2}$ | 42.5 $^{\uparrow 5.6}$ | 36.6 $^{\uparrow 2.9}$ | 34.2 $^{\uparrow 8.7}$ | 28.3 $^{\uparrow 14.0}$ | 36.3 $^{\uparrow 1.9}$ | 27.8 $^{\uparrow 3.7}$ |
| | FADRM | Soft-Only | 47.7 | 45.5 | 40.4 | 34.8 | 28.5 | 18.7 | 36.0 | 27.8 |
| | | Ours | 50.3 $^{\uparrow 2.6}$ | 49.5 $^{\uparrow 4.0}$ | 47.7 $^{\uparrow 7.3}$ | 42.7 $^{\uparrow 7.9}$ | 40.9 $^{\uparrow 12.4}$ | 36.9 $^{\uparrow 18.2}$ | 38.2 $^{\uparrow 2.2}$ | 30.7 $^{\uparrow 2.9}$ |

*Table 9.* Impact of different training schedules.

| Hard-Soft | Soft-Hard | Hard-Soft-Hard | Soft-Hard-Soft |
| --- | --- | --- | --- |
| 17.0 % | 14.2 % | 11.3 % | **35.6 %** |

fore the model has converged, disrupting the soft-to-hard transition and degrading soft–hard gradient alignment. The two failure modes ultimately reflect a single underlying trade-off between early-stage representation learning and late-stage semantic refinement, which we resolve by allocating the soft-label budget evenly across the two phases, consistent with our theoretical schedule.

*Table 10.* Effect of first and last soft-label phase durations.

| First Soft Duration | Last Soft Duration | Acc |
| --- | --- | --- |
| 50 | 100 | 35.2 |
| 100 | 50 | 34.7 |
| 75 | 75 | **35.6** |

**Effect of Label-Smoothing ($\alpha$).** As shown in Table 11, $\alpha = 0.8$ is optimal across generation methods. We attribute this to the fact that synthetic data carries less clear semantics than real data, so stronger label flattening is needed to produce smoother, higher-entropy targets for stable calibration, leading to more stable optimization.

*Table 11.* Effect of Label-Smoothing Rate.

| | 0.0 | 0.2 | 0.4 | 0.6 | 0.8 | 0.9 |
| --- | --- | --- | --- | --- | --- | --- |
| FADRM | 35.0 | 35.2 | 35.2 | 35.4 | **35.6** | 35.3 |
| LPLD | 26.9 | 27.3 | 27.5 | 27.9 | **30.0** | 27.9 |
| SRe²L | 29.6 | 30.0 | 30.9 | 31.5 | **31.9** | 31.3 |
| RDED | 21.9 | 22.9 | 23.2 | 24.0 | **24.4** | 23.8 |

## 4.4. Analysis

**LVSD Quantification.** To quantify LVSD across teacher models, we define the *LVSD ratio* $R(\tilde{x}) = \frac{\mathrm{Tr}(\hat{\Sigma}_{\mathrm{strong}})}{\mathrm{Tr}(\hat{\Sigma}_{\mathrm{weak}}) + \varepsilon}$, where the numerator captures prediction variance under *strong* (local-view) augmentations and the denominator under *weak* (global-view) augmentations. Thus, $R(\tilde{x})$ measures semantic drift between local and global views. As shown in Table 12, $R$ is consistently large across backbones, indicating pronounced LVSD and motivating semantic calibration under limited soft-label coverage.

*Table 12.* Quantitative analysis of Local-View Semantic Drift (LVSD) across teacher models. A larger $R$ indicates higher prediction variance under local views relative to global, confirming that LVSD is substantial across architectures.

| Teacher | Mean $\mathrm{Tr}(\hat{\Sigma}_{\mathrm{weak}})$ | Mean $\mathrm{Tr}(\hat{\Sigma}_{\mathrm{strong}})$ | $\log_{10}(\text{Mean } R)$ | $\mathbf{p}(R > 1)$ |
| --- | --- | --- | --- | --- |
| ResNet-18 | $4.84 \times 10^{-15}$ | 0.0102 | 3.27 | 97.2% |
| MobileNetV2 | $4.42 \times 10^{-15}$ | 0.3756 | 5.28 | 99.2% |
| ShuffleNetV2 | $4.51 \times 10^{-15}$ | 0.0591 | 5.35 | 98.0% |

**Semantic Calibration.** To verify that HALD mitigates semantic drift and improves generalization, we analyze crop-level consistency and prediction alignment against a reference model trained with full soft-label coverage. Crop-level consistency measures agreement across crops of the same image via Jensen–Shannon divergence and cosine similarity before and after Stage B, while prediction alignment assesses how closely student predictions (with/without hard calibration) match the reference on unseen data. As shown in Table 13, HALD improves both metrics, confirming the effectiveness of hard-label calibration in reducing semantic

drift and enhancing performance.

*Table 13.* **Top:** Crop-level consistency before and after Stage B. **Bottom:** Prediction alignment with a reference model trained under full soft-label coverage on unseen data.

| Stage | Cos. Sim. |
| --- | --- |
| Before Stage B | 0.74 |
| After Stage B | **0.96** |

| | Cos. Sim. |
| --- | --- |
| w/o Hard Calibration | 0.46 |
| w/ Hard Calibration | **0.62** |

**Cross-Architecture Generalization.** To assess backbone-agnostic effectiveness, we evaluate **HALD** across heterogeneous backbones, ranging from lightweight networks to larger architectures. As reported in Table 14, **HALD** yields consistent improvements across all backbones examined, demonstrating that the advantages of our method are not tied to specific architectural choices. For instance, **HALD** improves ShuffleNetV2 by +3.1%, indicating substantial gains even for compact models with limited capacity. These results collectively suggest that the benefits of our training paradigm are architecture-agnostic, robust to variations in network complexity, and scale favorably with parameter counts and model capacities.

*Table 14.* Results on cross-architecture generalization, showing Top-1 accuracy (%) with IPC=10 under SLC=100.

| Model | #Params | RDED | LPLD | FADRM | Ours |
| --- | --- | --- | --- | --- | --- |
| ResNet-18 | 11.7M | 19.9 | 23.1 | 27.9 | **35.6** $^{\uparrow 7.7}$ |
| ResNet-50 | 25.6M | 25.2 | 27.3 | 36.1 | **38.0** $^{\uparrow 1.9}$ |
| EfficientNet-B0 | 39.6M | 19.5 | 26.1 | 36.4 | **37.8** $^{\uparrow 1.4}$ |
| MobileNetV2 | 3.4M | 17.3 | 25.1 | 34.2 | **35.3** $^{\uparrow 1.1}$ |
| DenseNet121 | 8.0M | 28.3 | 36.7 | 43.3 | **44.3** $^{\uparrow 1.0}$ |
| ShuffleNetV2-0.5x | 1.4M | 17.9 | 21.1 | 29.2 | **32.3** $^{\uparrow 3.1}$ |
| Vit-Tiny | 13M | 3.2 | 3.8 | 5.6 | **8.9** $^{\uparrow 3.3}$ |
| VGG-11 | 133M | 26.3 | 28.9 | 31.0 | **33.6** $^{\uparrow 2.6}$ |
| VGG-16 | 138M | 28.9 | 34.3 | 36.2 | **37.4** $^{\uparrow 1.2}$ |

**Prediction Consistency with Teacher Model.** We compare the prediction consistency on unseen data between models trained with and without the final soft-label refinement stage, to empirically demonstrate the performance gains introduced by this phase. As shown in Table 15, prediction alignment with the teacher on unseen data improves notably after the final refinement stage.

*Table 15.* Comparison of prediction consistency with the teacher model on unseen data, w/ and w/o the final soft-label refinement.

| Method | JS Divergence | Cosine Similarity |
| --- | --- | --- |
| w/o final soft-label refinement | 0.61 | 19.8 |
| w/ final soft-label refinement | 0.38 | 43.8 |

**Conventional Large-Scale Dataset Experiment.** To evaluate generalization beyond synthetic data, we test **HALD** on a randomly sampled subset of ImageNet-1K. As shown in Table 16, hard-label supervision consistently improves performance, demonstrating robustness under realistic visual conditions and effective transfer to large-scale, real-world

benchmarks. These results highlight the practical applicability and reliability of **HALD** in real-data settings.

*Table 16. Soft-Only* vs. **HALD** on real dataset.

| ResNet-18 | SLC=100 (190 MB) | | SLC=50 (95 MB) | |
| --- | --- | --- | --- | --- |
| | *Soft-Only* | **Ours** | *Soft-Only* | **Ours** |
| IPC=10 | 29.9 | **34.4** | 26.9 | **28.6** |
| IPC=50 | 26.9 | **44.7** | 19.1 | **40.9** |

## 5. Visualization

To further illustrate the effect of hard label calibration, we visualize the learned feature representations via UMAP (McInnes et al., 2018) in Figure 4. Under Soft-Only training, the feature space exhibits a chaotic, entangled structure with poor inter-class separation (Silhouette = 0.162), reflecting the semantic inconsistency induced by limited soft-label coverage. In contrast, the model trained using **HALD** produces compact, well-separated clusters (Silhouette = 0.636), providing clear visual evidence that hard-label calibration effectively mitigates local semantic drift and yields significantly more discriminative representations.

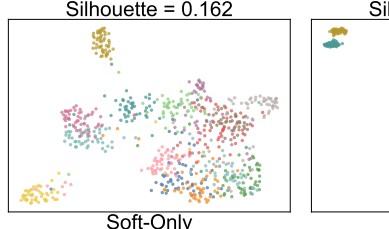 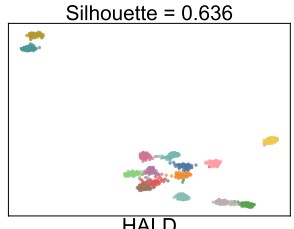

*Figure 4.* UMAP visualization of feature representations. Soft-Only yields entangled clusters, while **HALD** produces well-separated structures, confirming reduced semantic drift.

## 6. Conclusion

In this work, we revisited the limits of soft-label supervision in dataset distillation under tight storage budgets and identified *local semantic drift* as a core failure mode when only a few per-image crops (and thus soft labels) are retained. We showed theoretically that the expected objective mismatch between reduced-crop and sufficient-crop training admits a strictly positive lower bound that scales inversely with the number of crops, and we proved that combining soft and hard labels does not introduce gradient inconsistency. Building on these insights, we proposed a lightweight calibration paradigm **HALD**, where hard labels act as content-agnostic anchors that realign supervision while preserving the fine-grained benefits of soft labels. Experiments on large-scale settings (e.g., ImageNet-1K) demonstrate that **HALD** mitigates drift, improves generalization and robustness, and substantially reduces storage overhead, providing a practical path toward scalable distillation.

## Acknowledgments

This work is supported by the MBZUAI-WIS Joint Program for Artificial Intelligence Research.

## Impact Statement

This work improves the reliability and efficiency of dataset distillation by mitigating local semantic drift that can arise when soft-label supervision is sparse or crop-dependent. By reintroducing hard labels as a lightweight semantic anchor, our method can reduce reliance on storing massive teacher outputs, lowering label storage and energy costs and enabling broader access to distillation on hardware or devices. Potential risks include reinforcing noise or bias present in hard labels and over-calibrating toward dataset-specific system, we therefore encourage careful label checking, reporting of detailed performance, and ablations on label quality and class imbalance when deploying the approach.

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

# Appendix

## A. Notation

| Symbol | Definition |
|---|---|
| $\mathcal{O}$ | Original dataset of labeled samples |
| $\mathcal{C}$ | Distilled dataset (small synthetic set) |
| $(x, y)$ | Input sample $x$ with ground-truth label $y$ |
| $\tilde{x}, \tilde{y}$ | Synthetic (distilled) sample and label |
| $f_\theta$ | Model parameterized by $\theta$ |
| $\theta_\mathcal{O}, \theta_\mathcal{C}$ | Parameters trained on $\mathcal{O}$ or $\mathcal{C}$ |
| $\mathcal{L}(\cdot, \cdot)$ | Per-sample loss functional |
| $\mathcal{T}(\tilde{x})$ | Augmentation distribution of $\tilde{x}$ |
| $x^{(\text{crop})}$ | Random crop sampled from $\mathcal{T}(\tilde{x})$ |
| $\tilde{p}(x^{(\text{crop})})$ | Teacher soft prediction on a crop |
| $\bar{p}$ | Crop-averaged teacher prediction $\mathbb{E}[\tilde{p}(x^{(\text{crop})})]$ |
| $\Sigma$ | Covariance of teacher predictions across crops |
| $\hat{p}_s$ | Empirical average prediction over $s$ crops |
| SLC | Soft Labels per Class (storage budget) |
| $n_{\text{soft}}, n_{\text{hard}}, n_{\text{total}}$ | Epoch budgets for soft-/hard-label training |
| $\Omega_{\text{soft}}$ | Global pool of stored soft labels |
| $\Omega_{\text{cal}}$ | Calibration sampling space with hard labels |
| $\text{LS}_\alpha(\cdot)$ | Label smoothing distribution with ratio $\alpha$ |
| $t_{\lambda,\alpha}(y, y')$ | CutMix target between labels $y, y'$ |
| $q_\theta(\cdot \mid x)$ | Student predictive distribution on input $x$ |
| $\hat{\theta}_s^{\text{A}}, \hat{\theta}^{\text{B}}, \hat{\theta}$ | Parameters after Stages A, B, and C |
| $H_\star$ | Hessian of $\mathcal{L}_{\text{ideal}}$ at optimum |
| $\Sigma_\star$ | Gradient covariance at optimum |
| $s_{\text{eff}}$ | Effective sample size after calibration |

*Table 17.* List of common mathematical symbols used in this paper.

## B. Proof

### B.1. Lower Bound on the Empirical Loss Bias under Limited Crop Supervision

*Proof of Theorem 3.5.* Fix the synthetic image $\tilde{x}$ and its augmentation law $\mathcal{T}(\tilde{x})$. Define the per-crop loss random variable:

$$X := \mathcal{L}\left[\tilde{p}, q_\theta(\cdot \mid \tilde{x}^{(\text{crop})})\right], \quad \text{where } \tilde{x}^{(\text{crop})} \sim \mathcal{T}(\tilde{x}).$$

Let $\mu := \mathbb{E}X = \mathcal{L}_{\text{ideal}}(\theta; \tilde{x})$, $\sigma^2 := \text{Var}(X) < \infty$, and a finite kurtosis $\kappa := \frac{\mathbb{E}[(X-\mu)^4]}{\sigma^4} \in [1, \infty)$. For $s$ i.i.d. crops $(\tilde{x}_i^{(\text{crop})})_{i=1}^s$ drawn from $\mathcal{T}(\tilde{x})$, set:

$$X_i := \mathcal{L}\left[\tilde{p}_i, q_\theta(\cdot \mid \tilde{x}_i^{(\text{crop})})\right] \quad \text{(i.i.d. copies of } X\text{)}, \qquad \bar{X}_s := \frac{1}{s}\sum_{i=1}^s X_i = \mathcal{L}_s(\theta; \tilde{x}).$$

Our target quantity is $\mathbb{E}\left[|\bar{X}_s - \mu|\right] = \mathbb{E}\left[|\mathcal{L}_s(\theta; \tilde{x}) - \mathcal{L}_{\text{ideal}}(\theta; \tilde{x})|\right]$.

Step 1 (Second and fourth moments of the centered sample mean). Let $Y_i := X_i - \mu$ so that $\mathbb{E}Y_i = 0$, $\mathbb{E}Y_i^2 = \sigma^2$, and $\mathbb{E}Y_i^4 = \kappa \sigma^4$. Define the centered sample mean:

$$W := \bar{X}_s - \mu = \frac{1}{s}\sum_{i=1}^s Y_i.$$

By independence,

$$\mathbb{E}[W^2] \;=\; \frac{1}{s^2}\sum_{i=1}^{s}\mathbb{E}[Y_i^2] \;=\; \frac{\sigma^2}{s}.$$

For the fourth moment, only index patterns that are all equal or pairwise-equal contribute:

$$\mathbb{E}\left[\left(\sum_{i=1}^{s}Y_i\right)^4\right] \;=\; s\,\mathbb{E}[Y_1^4] \;+\; 3\,s(s-1)\,\sigma^4 \;=\; s\,\kappa\,\sigma^4 \;+\; 3s(s-1)\sigma^4.$$

Therefore,

$$\mathbb{E}[W^4] \;=\; \frac{1}{s^4}\,\mathbb{E}\left[\left(\sum_{i=1}^{s}Y_i\right)^4\right] \;=\; \frac{\sigma^4}{s^3}\big(\kappa + 3(s-1)\big).$$

**Step 2 (Paley–Zygmund on $W^2$).** Let $Z := W^2 \geq 0$. For any $\theta \in (0,1)$, Paley–Zygmund yields:

$$\mathbb{P}(Z \geq \theta\,\mathbb{E}Z) \;\geq\; (1-\theta)^2\,\frac{(\mathbb{E}Z)^2}{\mathbb{E}[Z^2]}.$$

Using $\mathbb{E}Z = \mathbb{E}[W^2] = \sigma^2/s$ and $\mathbb{E}[Z^2] = \mathbb{E}[W^4] = \frac{\sigma^4}{s^3}\big(\kappa + 3(s-1)\big)$ from Step 1,

$$\mathbb{P}\left(W^2 \geq \theta\,\frac{\sigma^2}{s}\right) \;\geq\; (1-\theta)^2\,\frac{\left(\frac{\sigma^2}{s}\right)^2}{\frac{\sigma^4}{s^3}\big(\kappa + 3(s-1)\big)} \;=\; (1-\theta)^2 \cdot \frac{s}{\kappa + 3(s-1)}.$$

**Step 3 (From a small-ball event to a first-moment bound).** For any $t > 0$, $\mathbb{E}|W| \geq t\,\mathbb{P}(|W| \geq t)$. Choose $t := \sqrt{\theta\,\mathbb{E}W^2} = \frac{\sigma}{\sqrt{s}}\sqrt{\theta}$ to match Step 2. Then:

$$\mathbb{E}|W| \;\geq\; \frac{\sigma}{\sqrt{s}}\sqrt{\theta}\,\mathbb{P}\left(W^2 \geq \theta\,\frac{\sigma^2}{s}\right) \;\geq\; \frac{\sigma}{\sqrt{s}}\sqrt{\theta}\,(1-\theta)^2\,\frac{s}{\kappa + 3(s-1)}.$$

**Step 4 (Optimize $\theta$).** Define $g(\theta) := \sqrt{\theta}\,(1-\theta)^2$ on $\theta \in [0,1]$. A direct derivative check gives $g'(\theta) = 0$ at $\theta^\star = \frac{1}{5}$ and $g(\theta^\star) = \frac{16}{25\sqrt{5}}$. Plugging $\theta^\star$ into Step 3 yields

$$\mathbb{E}|W| \;\geq\; \frac{\sigma}{\sqrt{s}} \cdot \frac{16}{25\sqrt{5}} \cdot \frac{s}{\kappa + 3(s-1)}.$$

**Step 5 (Uniform-in-$s$ simplification).** Consider $h(s) := \dfrac{s}{\kappa + 3(s-1)} = \dfrac{s}{3s + \kappa - 3}$ for $s \geq 1$. Then

$$h'(s) = \frac{(\kappa - 3)}{(\kappa + 3s - 3)^2}.$$

Hence:

- If $\kappa \geq 3$, $h$ is nondecreasing on $[1,\infty)$, so $\min_{s\geq 1} h(s) = h(1) = \frac{1}{\kappa} \leq \frac{1}{3}$.

- If $\kappa < 3$, $h$ is strictly decreasing and $\inf_{s\geq 1} h(s) = \lim_{s\to\infty} h(s) = \frac{1}{3}$, while $h(1) = \frac{1}{\kappa} > \frac{1}{3}$.

In both cases,

$$\frac{s}{\kappa + 3(s-1)} \;\geq\; \min\left\{\frac{1}{\kappa}, \frac{1}{3}\right\}.$$

Combining with the bound above concludes

$$\mathbb{E}\big[\big|\bar{X}_s - \mu\big|\big] \;=\; \mathbb{E}\big[\big|\mathcal{L}_s(\theta; \tilde{x}) - \mathcal{L}_{\mathrm{ideal}}(\theta; \tilde{x})\big|\big] \;\geq\; \frac{\sigma}{\sqrt{s}} \cdot \frac{16}{25\sqrt{5}} \cdot \min\left\{\frac{1}{\kappa}, \frac{1}{3}\right\}.$$

This is exactly the claimed bound. $\qquad\square$

## B.2. Limited Crop Supervision Degrades Generalization Performance

**Interpretation of Assumptions (A1–A5).** To establish a finite-sample lower bound on the excess population risk of empirical risk minimization (ERM), we adopt five standard assumptions that ensure local regularity and statistical stability near the population minimizer $\hat{\theta}_\star$. Below we provide an intuitive interpretation of each:

- **(A1) Unbiased Score and Covariance.** We assume $\mathbb{E}[g(\hat{\theta}_\star; x)] = 0$ and that the covariance $\Sigma_\star = \mathrm{Cov}(g(\hat{\theta}_\star; x))$ exists with bounded $(2 + \kappa)$-moment. This ensures that the gradient noise at the optimum is well-behaved, and the matrix $\Sigma_\star$ characterizes the first-order variance that drives the leading term in the excess risk.

- **(A2) Hessian Lipschitz Continuity.** The population loss Hessian is assumed to be $L_H$-Lipschitz in a neighborhood of $\hat{\theta}_\star$. This smoothness enables accurate control over second-order Taylor expansions and guarantees that local quadratic approximations remain valid.

- **(A3) Local Uniform Concentration.** We require that both the empirical Hessian and empirical gradient fluctuations concentrate uniformly to their population counterparts in a neighborhood of $\hat{\theta}_\star$, with deviations decaying as $O(1/\sqrt{s})$. This ensures that the empirical loss landscape closely tracks the population landscape, which is essential for Newton-type approximations and influence-function expansions.

- **(A4) ERM Stays Local.** With high probability, the empirical minimizer $\hat{\theta}_s$ lies within a fixed ball around $\hat{\theta}_\star$. This ensures that our analysis can be restricted to a well-behaved local region where smoothness and concentration assumptions hold.

- **(A5) Bounded Loss Near Optimum.** The population loss is assumed to be uniformly bounded within the neighborhood of interest. This provides worst-case control when $\hat{\theta}_s$ falls outside the local region, allowing us to bound the risk in rare failure cases.

Together, these assumptions provide a sufficient foundation to develop second-order expansions around $\hat{\theta}_\star$, rigorously control deviation terms, and derive a tight lower bound on the expected excess risk of finite-sample ERM. Next, we will formally prove the theorem.

*Proof of Theorem 3.6.* Step 0 (Good vs. bad events). Define the event:

$$\mathcal{E}_s := \left\{ \text{(A3) holds and } \hat{\theta}_s \in \mathbb{B}(\hat{\theta}_\star, r_0) \right\}.$$

By (A3)–(A4), $\Pr(\mathcal{E}_s) \geq 1 - \delta'_s$, with $\delta'_s \leq \delta_s + 2\delta$, where $\delta$ can be chosen polynomially small (e.g. $\delta = s^{-3}$). Split the expectation:

$$\mathbb{E}\left[ \mathcal{L}_{\mathrm{ideal}}(\hat{\theta}_s) - \mathcal{L}_{\mathrm{ideal}}(\hat{\theta}_\star) \right] = \mathbb{E}[\cdot; \mathcal{E}_s] + \mathbb{E}[\cdot; \mathcal{E}_s^c].$$

On the bad event, (A5) implies $\mathcal{L}_{\mathrm{ideal}}(\hat{\theta}_s) - \mathcal{L}_{\mathrm{ideal}}(\hat{\theta}_\star) \geq -B$, so $\mathbb{E}[\cdot; \mathcal{E}_s^c] \geq -C_b\, \delta_s$. Thus it suffices to prove the stated bound conditional on $\mathcal{E}_s$.

Step 1 (Influence-function expansion). Let $U_s(\theta) := \nabla \mathcal{L}_s(\theta) - \nabla \mathcal{L}_{\mathrm{ideal}}(\theta)$, $\bar{g}_s := U_s(\hat{\theta}_\star) = \frac{1}{s} \sum_{i=1}^{s} g(\hat{\theta}_\star; x_i)$. On $\mathcal{E}_s$, Lipschitz continuity yields $\|U_s(\theta) - U_s(\hat{\theta}_\star)\| \leq \bar{c}\, s^{-1/2} \|\theta - \hat{\theta}_\star\|$. Using the Newton map $T_s(\theta) := \theta - H_\star^{-1} \nabla \mathcal{L}_s(\theta)$, and setting $r_s := c\, \|\bar{g}_s\|$ with $c > 0$ depending only on $(\mu, L_H, \bar{c})$, one shows that $T_s$ is a contraction mapping $\mathbb{B}(\hat{\theta}_\star, r_s)$ into itself, with unique fixed point $\hat{\theta}_s$. Define $\Delta_s := \hat{\theta}_s - \hat{\theta}_\star$. Then:

$$\Delta_s = -H_\star^{-1} \bar{g}_s + R_s,$$

where the remainder satisfies $\|R_s\| \lesssim \|\bar{g}_s\|^2 + s^{-1/2} \|\bar{g}_s\|$. Since $\mathbb{E}[\|\bar{g}_s\|] = O(s^{-1/2})$ and $\mathbb{E}[\|\bar{g}_s\|^2] = O(s^{-1})$, it follows that

$$\mathbb{E}[\|R_s\| \mid \mathcal{E}_s] = O(s^{-1}). \tag{6}$$

Step 2 (Quadratic term). With $\|v\|_{H_\star}^2 := v^\top H_\star v$, one has

$$\|\Delta_s\|_{H_\star}^2 = \|H_\star^{-1} \bar{g}_s\|_{H_\star}^2 + 2\langle H_\star^{-1} \bar{g}_s, R_s \rangle_{H_\star} + \|R_s\|_{H_\star}^2.$$

Taking expectations conditional on $\mathcal{E}_s$ and using (6),

$$\mathbb{E}[\|\Delta_s\|_{H_\star}^2 \mid \mathcal{E}_s] = \frac{1}{s}\operatorname{tr}(H_\star^{-1}\Sigma_\star) + O(s^{-3/2}).$$

**Step 3 (Excess risk).** The integral second-order expansion gives

$$\mathcal{L}_{\text{ideal}}(\hat{\theta}_s) - \mathcal{L}_{\text{ideal}}(\hat{\theta}_\star) = \tfrac{1}{2}\|\Delta_s\|_{H_\star}^2 + R_s^{(3)},$$

where $R_s^{(3)} := \int_0^1 (1-t)\,\Delta_s^\top(\nabla^2\mathcal{L}_{\text{ideal}}(\hat{\theta}_\star + t\Delta_s) - H_\star)\Delta_s\,dt$. By (A2), $|R_s^{(3)}| \leq (L_H/6)\|\Delta_s\|^3$. Since $\mathbb{E}\|\Delta_s\|^3 = O(s^{-3/2})$, it follows that

$$\mathbb{E}[R_s^{(3)} \mid \mathcal{E}_s] \geq -O(s^{-3/2}).$$

Therefore,

$$\mathbb{E}\big[\mathcal{L}_{\text{ideal}}(\hat{\theta}_s) - \mathcal{L}_{\text{ideal}}(\hat{\theta}_\star) \mid \mathcal{E}_s\big] \geq \frac{1}{2s}\operatorname{tr}(H_\star^{-1}\Sigma_\star) - \frac{C_1}{s^{3/2}} - \frac{C_2}{s^2}.$$

**Step 4 (Combine events).** Adding the contribution from $\mathcal{E}_s^{\mathrm{c}}$ gives

$$\mathbb{E}\big[\mathcal{L}_{\text{ideal}}(\hat{\theta}_s) - \mathcal{L}_{\text{ideal}}(\hat{\theta}_\star)\big] \geq \frac{1}{2s}\operatorname{tr}(H_\star^{-1}\Sigma_\star) - \frac{C_1}{s^{3/2}} - \frac{C_2}{s^2} - C_b\,\delta_s,$$

as claimed. $\qquad\square$

### B.3. Optimization Stability

#### B.3.1. PRELIMINARIES

**Lemma B.1** (Mixing bound via dual norms, Proof in Appendix B.3.1)**.** *Let $p, q \in \Delta^C$ and $g_1, \ldots, g_C \in \mathbb{R}^d$. Fix any norm $\|\cdot\|$ on $\mathbb{R}^d$ with dual norm $\|\cdot\|_*$, and define the diameter*

$$D := \sup_{i,j}\|g_i - g_j\| < \infty.$$

*Then*

$$\Big\|\sum_{c=1}^C (p_c - q_c)\,g_c\Big\| \leq \frac{D}{2}\,\|p - q\|_1.$$

*Moreover, the constant $D/2$ is optimal (tight when $C = 2$, $p = (1,0)$, $q = (0,1)$, $\|g_1 - g_2\| = D$).*

*Proof of Lemma B.1.* Write $r := p - q$ and note $\sum_c r_c = 0$. By the dual representation of the norm,

$$\Big\|\sum_c r_c g_c\Big\| = \sup_{\|u\|_* \leq 1}\Big\langle u, \sum_c r_c g_c\Big\rangle = \sup_{\|u\|_* \leq 1}\sum_c r_c\,\phi_c, \quad \text{where } \phi_c := \langle u, g_c\rangle.$$

Split the indices into $P := \{i : r_i > 0\}$ and $N := \{j : r_j < 0\}$, and let

$$T := \sum_{i \in P} r_i = \sum_{j \in N}|r_j| = \tfrac{1}{2}\|r\|_1 = \tfrac{1}{2}\|p - q\|_1.$$

Then for any fixed $u$,

$$\sum_c r_c\phi_c = \sum_{i \in P} r_i\phi_i - \sum_{j \in N}|r_j|\phi_j \leq \Big(\max_c \phi_c - \min_c \phi_c\Big)T,$$

because the linear form is maximized by assigning all positive mass to an index attaining $\max_c \phi_c$ and all negative mass to one attaining $\min_c \phi_c$. Hence

$$\Big\|\sum_c r_c g_c\Big\| \leq T\sup_{\|u\|_* \leq 1}\big(\max_c \phi_c - \min_c \phi_c\big) \leq T\sup_{\|u\|_* \leq 1}\sup_{i,j}|\phi_i - \phi_j|.$$

Finally,

$$|\phi_i - \phi_j| = |\langle u, g_i - g_j \rangle| \le \|u\|_* \|g_i - g_j\| \le \|g_i - g_j\|,$$

so $\sup_{\|u\|_* \le 1} \sup_{i,j} |\phi_i - \phi_j| \le \sup_{i,j} \|g_i - g_j\| = D$, and thus

$$\left\| \sum_c (p_c - q_c) g_c \right\| \le T D = \frac{D}{2} \|p - q\|_1.$$

For tightness, take $C = 2$, $p = (1, 0)$, $q = (0, 1)$; then $T = 1$, and choosing $g_1, g_2$ with $\|g_1 - g_2\| = D$ yields equality. $\square$

### B.3.2. FORMAL PROOF

*Proof of Theorem 3.7.* **Step 1 (Mixing stability).** By Lemma B.1, for any $p, q \in \Delta^C$,

$$\left\| \sum_{c=1}^C (p_c - q_c) g_c \right\| \le \frac{D}{2} \|p - q\|_1,$$

and the constant $\frac{D}{2}$ is tight (e.g. $C = 2$, $p = (1, 0)$, $q = (0, 1)$, $\|g_1 - g_2\| = D$).

**Step 2 (From differences to cosine).** Let

$$a := \nabla_\theta \mathcal{L}_{\text{soft}} = -\sum_c \tilde{p}_c g_c, \qquad b := \nabla_\theta \mathcal{L}_{\text{hard}} = -\sum_c \bar{p}_c^{(\alpha)} g_c.$$

For nonzero $a, b$, write $\hat{a} := a/\|a\|$ and $\hat{b} := b/\|b\|$. Then

$$1 - \cos(a, b) = \tfrac{1}{2}\|\hat{a} - \hat{b}\|^2 \le \|\hat{a} - \hat{b}\| = \left\| \frac{a}{\|a\|} - \frac{b}{\|b\|} \right\| \le \frac{\|a - b\|}{\|a\|} + \frac{\|a - b\|}{\|b\|} \le \frac{2\|a - b\|}{\min\{\|a\|, \|b\|\}}.$$

By the theorem's non-degeneracy assumption, $\min\{\|a\|, \|b\|\} \ge m_0 > 0$, hence

$$1 - \cos(a, b) \le \frac{2}{m_0} \|a - b\|.$$

Applying Step 1 with $p = \tilde{p}$ and $q = \bar{p}^{(\alpha)}$ yields

$$1 - \cos(a, b) \le \frac{2}{m_0} \cdot \frac{D}{2} \|\tilde{p} - \bar{p}^{(\alpha)}\|_1 = \frac{D}{m_0} \|\tilde{p} - \bar{p}^{(\alpha)}\|_1. \tag{7}$$

**Step 3 (Upper-bounding $\|\tilde{p} - \bar{p}^{(\alpha)}\|_1$).** Let $y = \arg\max_c \tilde{p}_c$ and $e_y$ be the one-hot at $y$. By the triangle inequality,

$$\|\tilde{p} - \bar{p}^{(\alpha)}\|_1 \le \|\tilde{p} - e_y\|_1 + \|e_y - \bar{p}^{(\alpha)}\|_1.$$

The two terms are exact:

$$\|\tilde{p} - e_y\|_1 = \sum_{c \ne y} \tilde{p}_c + |1 - \tilde{p}_y| = 2(1 - p_{\max}), \quad p_{\max} := \max_c \tilde{p}_c,$$

and

$$\|e_y - \bar{p}^{(\alpha)}\|_1 = \sum_{c \ne y} \frac{\alpha}{C} + \left| 1 - \left( 1 - \alpha + \frac{\alpha}{C} \right) \right| = 2\alpha \left( 1 - \frac{1}{C} \right).$$

Therefore

$$\|\tilde{p} - \bar{p}^{(\alpha)}\|_1 \le 2(1 - p_{\max}) + 2\alpha \left( 1 - \frac{1}{C} \right). \tag{8}$$

**Step 4 (Relating $1 - p_{\max}$ to entropy and rewriting via teacher entropy).** Using the standard inequality (natural logarithm),

$$H(\tilde{p}) \ge -\log p_{\max} \implies p_{\max} \ge e^{-H(\tilde{p})} \implies 1 - p_{\max} \le 1 - e^{-H(\tilde{p})} \le H(\tilde{p}),$$

where the last step uses $1 - e^{-x} \leq x$ for $x \geq 0$. Substituting (8) into (7) gives, for each crop,

$$1 - \cos(\nabla_\theta \mathcal{L}_{\text{soft}}, \nabla_\theta \mathcal{L}_{\text{hard}}) \; \leq \; \frac{D}{m_0}\Big\{2\big(1 - e^{-H(\tilde{p})}\big) + 2\alpha\big(1 - \tfrac{1}{C}\big)\Big\} \; \leq \; \frac{D}{m_0}\Big\{2H(\tilde{p}) + 2\alpha\big(1 - \tfrac{1}{C}\big)\Big\}.$$

Taking expectation over the crop distribution $\mathcal{T}(\tilde{x})$ (conditioning on the base image $\tilde{x}$), and introducing the notation

$$\mathsf{H}_{\text{teacher}}(\tilde{x}) := \mathbb{E}_{x^{(\text{crop})} \sim \mathcal{T}(\tilde{x})}\Big[ H\big(\tilde{p}(\cdot \mid x^{(\text{crop})})\big) \Big],$$

we obtain

$$\mathbb{E}_{\text{crop}}[\cos(\nabla_\theta \mathcal{L}_{\text{soft}}, \nabla_\theta \mathcal{L}_{\text{hard}})] \; \geq \; 1 \; - \; \frac{D}{m_0} \cdot \Big( 2\,\mathsf{H}_{\text{teacher}}(\tilde{x}) \; + \; 2\alpha\Big(1 - \frac{1}{C}\Big)\Big),$$

where the bracketed term can be viewed as a data-dependent alignment constant. That is, we may write

$$\mathbb{E}_{\text{crop}}[\cos(\nabla_\theta \mathcal{L}_{\text{soft}}, \nabla_\theta \mathcal{L}_{\text{hard}})] \; \geq \; 1 \; - \; \frac{D}{m_0} \cdot C_{\text{align}}(\tilde{x}, \alpha),$$

where

$$C_{\text{align}}(\tilde{x}, \alpha) := 2\,\mathsf{H}_{\text{teacher}}(\tilde{x}) \; + \; 2\alpha\Big(1 - \frac{1}{C}\Big).$$

$\square$

## B.4. HALD increases generalization performance

*Proof of Corollary 3.8.* Consider the scalar control–variate residual $r_\beta := u - \beta v$, $\beta \in \mathbb{R}$. Using linearity of expectation and $\|u\| = \|v\| = 1$,
$$\mathbb{E}\|r_\beta\|^2 = \mathbb{E}\|u\|^2 - 2\beta\,\mathbb{E}\langle u, v\rangle + \beta^2\mathbb{E}\|v\|^2 = 1 - 2\beta\,\mathbb{E}\langle u, v\rangle + \beta^2.$$
This quadratic is minimized at $\beta^\star = \mathbb{E}\langle u, v\rangle$, yielding

$$\min_\beta \mathbb{E}\|u - \beta v\|^2 = 1 - \big(\mathbb{E}\langle u, v\rangle\big)^2 \; \leq \; 1 - \rho_\star^2. \tag{1}$$

Center the residual $\tilde{r} := r_{\beta^\star} - \mathbb{E}[r_{\beta^\star}]$ so that $\mathbb{E}[\tilde{r}] = 0$. For i.i.d. copies $(\tilde{r}_i)_{i=1}^s$,

$$\mathbb{E}\Big\|\frac{1}{s}\sum_{i=1}^s \tilde{r}_i\Big\|^2 = \frac{1}{s^2}\sum_{i=1}^s \mathbb{E}\|\tilde{r}_i\|^2 = \frac{1}{s}\,\mathbb{E}\|\tilde{r}\|^2 \; \leq \; \frac{1}{s}\,\mathbb{E}\|r_{\beta^\star}\|^2 \; \leq \; \frac{1 - \rho_\star^2}{s},$$

where we used independence, zero mean (cross terms vanish), and (1). Interpreting the factor $(1 - \rho_\star^2)$ as variance contraction of the single-sample noise implies the same mean–square error as having $s_{\text{eff}}$ baseline samples with no contraction:

$$\frac{1 - \rho_\star^2}{s} = \frac{1}{s_{\text{eff}}} \quad \Longrightarrow \quad s_{\text{eff}} = \frac{s}{1 - \rho_\star^2}.$$

Because we used only a *scalar* control variate in the *direction* $v$, any richer use of the hard information (e.g., including magnitudes or conditional expectations) can only further reduce the left-hand side, hence the stated inequality $s_{\text{eff}} \geq s/(1 - \rho_\star^2)$. $\square$

*Insight.* Corollary 3.8 extends Theorem 3.7 by transforming gradient alignment into a formal variance-reduction guarantee. While Theorem 3.7 establishes that the soft-to-hard switch is optimization-coherent, Corollary 3.8 quantifies its benefit: stronger alignment between soft- and hard-label gradients ($\rho_\star > 0$) effectively increases the usable supervision by enlarging the effective sample size,

$$s_{\text{eff}} \geq \frac{s}{1 - \rho_\star^2}.$$

This shows that during the hard-label calibration stage, variance is reduced and semantic drift is corrected, providing the theoretical basis for the subsequent soft-label refinement that restores fine-grained teacher consistency on top of the variance-reduced representation.

## C. More Experimental Results

To more comprehensively evaluate HALD's performance, we present additional results for IPC=30 and IPC=40 in Table 18, where consistent improvements can be observed.

*Table 18.* Comprehensive ablation of the impact of incorporating hard-label supervision across state-of-the-art dataset distillation methods on ImageNet-1K and Tiny-ImageNet. All models are trained for 300 epochs under identical hyperparameters, with the evaluation protocol being the sole difference. $^{\dagger}$ denotes values reported by the corresponding original sources.

| IPC | Generation | Evaluation | ImageNet-1K | | | | | | Tiny-ImageNet | |
| --- | --- | --- | --- | --- | --- | --- | --- | --- | --- | --- |
| | | | SLC=300 | SLC=250 | SLC=200 | SLC=150 | SLC=100 | SLC=50 | SLC=100 | SLC=50 |
| IPC=30 | SRe$^2$L | Soft-Only | 40.1 | 38.6 | 34.1 | 30.9 | 23.5 | 9.8 | 31.0 | 19.5 |
| | | **Ours** | 43.7 $\uparrow^{3.6}$ | 42.8 $\uparrow^{4.2}$ | 40.9 $\uparrow^{6.8}$ | 39.6 $\uparrow^{8.7}$ | 35.8 $\uparrow^{12.3}$ | 20.0 $\uparrow^{13.9}$ | 32.5 $\uparrow^{1.5}$ | 23.6 $\uparrow^{4.1}$ |
| | RDED | Soft-Only | 37.2 | 34.0 | 30.2 | 25.9 | 21.1 | 7.9 | 27.8 | 15.8 |
| | | **Ours** | 42.5 $\uparrow^{5.3}$ | 41.1 $\uparrow^{7.1}$ | 39.5 $\uparrow^{9.3}$ | 39.4 $\uparrow^{13.5}$ | 35.0 $\uparrow^{13.9}$ | 23.5 $\uparrow^{15.6}$ | 32.9 $\uparrow^{5.1}$ | 26.7 $\uparrow^{10.9}$ |
| | LPLD | Soft-Only | 42.0 | 38.9 | 34.6 | 31.9 | 21.6 | 8.6 | 32.2 | 17.9 |
| | | **Ours** | 44.9 $\uparrow^{2.9}$ | 44.3 $\uparrow^{5.4}$ | 42.1 $\uparrow^{7.5}$ | 40.9 $\uparrow^{9.0}$ | 36.1 $\uparrow^{14.5}$ | 22.7 $\uparrow^{14.1}$ | 35.2 $\uparrow^{3.0}$ | 24.6 $\uparrow^{6.7}$ |
| | FADRM | Soft-Only | 46.6 | 44.2 | 39.2 | 37.1 | 26.9 | 10.9 | 36.4 | 23.5 |
| | | **Ours** | 50.7 $\uparrow^{4.1}$ | 49.2 $\uparrow^{5.0}$ | 47.3 $\uparrow^{8.1}$ | 46.5 $\uparrow^{9.4}$ | 43.5 $\uparrow^{16.6}$ | 29.3 $\uparrow^{18.4}$ | 38.3 $\uparrow^{1.9}$ | 29.6 $\uparrow^{6.1}$ |
| IPC=40 | SRe$^2$L | Soft-Only | 39.8 | 38.2 | 35.1 | 29.0 | 20.4 | 11.7 | 30.4 | 21.0 |
| | | **Ours** | 44.9 $\uparrow^{5.1}$ | 44.4 $\uparrow^{5.2}$ | 43.3 $\uparrow^{8.2}$ | 39.3 $\uparrow^{10.3}$ | 35.0 $\uparrow^{14.6}$ | 26.9 $\uparrow^{15.2}$ | 31.5 $\uparrow^{1.1}$ | 24.0 $\uparrow^{3.0}$ |
| | RDED | Soft-Only | 36.6 | 34.3 | 32.1 | 25.5 | 19.0 | 11.0 | 26.8 | 19.7 |
| | | **Ours** | 44.5 $\uparrow^{7.9}$ | 43.2 $\uparrow^{8.9}$ | 42.1 $\uparrow^{10.0}$ | 38.5 $\uparrow^{13.0}$ | 35.3 $\uparrow^{16.3}$ | 28.2 $\uparrow^{17.2}$ | 31.5 $\uparrow^{4.7}$ | 24.5 $\uparrow^{4.8}$ |
| | LPLD | Soft-Only | 40.6 | 39.1 | 36.3 | 27.7 | 21.1 | 12.9 | 29.3 | 20.3 |
| | | **Ours** | 45.7 $\uparrow^{5.1}$ | 44.6 $\uparrow^{5.5}$ | 43.9 $\uparrow^{7.6}$ | 40.2 $\uparrow^{12.5}$ | 35.1 $\uparrow^{14.0}$ | 26.6 $\uparrow^{13.7}$ | 31.7 $\uparrow^{2.4}$ | 23.6 $\uparrow^{3.3}$ |
| | FADRM | Soft-Only | 46.1 | 45.0 | 41.4 | 31.2 | 22.8 | 14.0 | 34.1 | 27.0 |
| | | **Ours** | 51.2 $\uparrow^{5.1}$ | 51.0 $\uparrow^{6.0}$ | 48.7 $\uparrow^{7.3}$ | 45.9 $\uparrow^{14.7}$ | 40.8 $\uparrow^{20.0}$ | 32.9 $\uparrow^{18.9}$ | 35.9 $\uparrow^{1.8}$ | 30.1 $\uparrow^{3.1}$ |

## D. Adaptive Phase-Switching Mechanism

The fixed phase schedule requires specifying the soft-label convergence length $n_{\text{soft}}$ in advance, which may demand dataset-specific tuning. Here we describe an adaptive variant that eliminates this requirement via two theory-guided proxies, one for each phase transition.

**Phase 0→1 (Soft→Hard).** The first transition fires when soft-label training has sufficiently stabilized, i.e., when further soft-label epochs yield diminishing returns. Concretely, we monitor the validation accuracy over a sliding window of $w$ epochs and switch to Stage B once the improvement falls below a threshold $\delta_{\text{acc}}$:

$$\Delta_{\text{acc}}^{(t)} = \frac{1}{w} \sum_{i=0}^{w-1} \left( \text{Acc}^{(t-i)} - \text{Acc}^{(t-i-1)} \right) < \delta_{\text{acc}}. \tag{9}$$

This criterion is theoretically grounded: by Theorem 3.7, the soft-to-hard gradient alignment improves as training converges, so a plateau in validation accuracy serves as a reliable proxy for the alignment strength required by the Soft→Hard transition.

**Phase 1→2 (Hard→Soft).** The second transition fires when hard-label calibration saturates, i.e., when the student's predictive distribution ceases to change meaningfully. We track the normalized student entropy

$$\tilde{H}^{(t)} = \frac{1}{|\mathcal{B}|} \sum_{x \in \mathcal{B}} \frac{H\left( q_{\theta^{(t)}}(\cdot \mid x) \right)}{\log C}, \tag{10}$$

where $C$ is the number of classes, so that $\tilde{H}^{(t)} \in [0, 1]$ regardless of dataset scale, making the trigger directly comparable across datasets without additional tuning. The transition to Stage C is triggered when the entropy plateau condition

$$\left| \tilde{H}^{(t)} - \tilde{H}^{(t-w)} \right| < \delta_H \tag{11}$$

is satisfied. This is consistent with Corollary 3.8: once the effective sample size $s_{\text{eff}}$ ceases to grow, additional hard-label epochs provide no further variance reduction, and returning to soft-label refinement recovers fine-grained teacher semantics.

**Empirical validation.** Table 19 compares the adaptive and fixed schedules. The adaptive schedule matches the fixed one, confirming that the two theory-guided proxies reliably recover the manually tuned schedule without dataset-specific hyperparameter search.

*Table 19.* Top-1 accuracy (%) of HALD with adaptive vs. fixed phase scheduling.

| Dataset | HALD (adaptive) | HALD (fixed) |
|---|---|---|
| CIFAR-100 | 25.8 | 26.0 |
| ImageNet-1K | 35.2 | 35.6 |

# E. Generation Method

### E.1. SRe2L

SRe$^2$L (Yin et al., 2023) decouples dataset condensation into three stages, *Squeeze*, *Recover*, and *Relabel*, so that the model training on $\mathcal{O}$ and the optimization of $\mathcal{C}$ never interleave. Concretely:

**Stage I: Squeeze (train on $\mathcal{O}$).** Learn a reference model by standard ERM on the original data:

$$\theta_{\mathcal{O}} \;=\; \arg\min_{\theta}\; \mathbb{E}_{(x,y)\sim\mathcal{O}}\big[\,\mathcal{L}\big(f_\theta(x),\,y\big)\,\big].$$

**Stage II: Recover (optimize $\tilde{x}$ with BN-consistency and classification).** Fix $f_{\theta_{\mathcal{O}}}$ and optimize the images $\tilde{x} \in \mathcal{C}$ (labels $y$ are class indices) by matching both the classifier head and global BatchNorm (BN) statistics accumulated on $\mathcal{O}$. With random crops $x^{(\mathrm{crop})} \sim \mathcal{T}(\tilde{x})$,

$$\min_{\tilde{x}\in\mathcal{C}} \;\underbrace{\mathbb{E}_{x^{(\mathrm{crop})}\sim\mathcal{T}(\tilde{x})}\big[\mathcal{L}\big(f_{\theta_{\mathcal{O}}}\big(x^{(\mathrm{crop})}\big),\,y\big)\big]}_{\text{classification (single-level)}} \;+\; \alpha_{\mathrm{BN}} \;\underbrace{R_{\mathrm{BN}}(\tilde{x})}_{\text{BN-consistency}} \;+\; \alpha_{\ell_2}\,\|\tilde{x}\|_2^2 \;+\; \alpha_{\mathrm{TV}}\,\mathrm{TV}(\tilde{x}).$$

The BN-consistency regularizer matches per-layer running mean/variance of $f_{\theta_{\mathcal{O}}}$:

$$R_{\mathrm{BN}}(\tilde{x}) = \sum_{\ell} \big\|\mu_\ell(x^{(\mathrm{crop})}) - \mathrm{BNRM}_\ell\big\|_2^2 + \sum_{\ell} \big\|\sigma_\ell^2(x^{(\mathrm{crop})}) - \mathrm{BNRV}_\ell\big\|_2^2,$$

where $\mathrm{BNRM}_\ell, \mathrm{BNRV}_\ell$ are the global running mean/variance stored in the $\ell$-th BN of $f_{\theta_{\mathcal{O}}}$. Multi-crop optimization (sampling $x^{(\mathrm{crop})}$ repeatedly from $\mathcal{T}(\tilde{x})$) enriches local semantics and constrains updates to the cropped region, which empirically improves recovery.

**Stage III: Relabel (crop-level soft labels and student training).** For each recovered $\tilde{x}$, draw $s$ crops $x_i^{(\mathrm{crop})} \sim \mathcal{T}(\tilde{x})$ and obtain teacher soft predictions

$$\tilde{p}\Big(x_i^{(\mathrm{crop})}\Big) \;=\; q_{\theta_{\mathcal{O}}}\Big(\cdot \,\Big|\, x_i^{(\mathrm{crop})}\Big), \qquad \bar{p} = \mathbb{E}\big[\tilde{p}(x^{(\mathrm{crop})})\big], \qquad \hat{p}_s = \tfrac{1}{s}\sum_{i=1}^{s} \tilde{p}\Big(x_i^{(\mathrm{crop})}\Big), \tag{L1}$$

and optionally characterize crop-prediction variability by $\Sigma = \mathrm{Cov}\big(\tilde{p}(x^{(\mathrm{crop})})\big)$. Train a student on $\mathcal{C}$ with crop-level distillation (temperature $\tau$):

$$\min_{\theta} \; \mathbb{E}_{\tilde{x}\in\mathcal{C}}\left[\frac{1}{s}\sum_{i=1}^{s} \mathrm{CE}\Big(\mathrm{softmax}(\tfrac{1}{\tau}\,\tilde{p}(x_i^{(\mathrm{crop})})),\; q_\theta\Big(\cdot \,\Big|\, x_i^{(\mathrm{crop})}\Big)\Big)\right]. \tag{L2}$$

However, solely aligning to global running statistics induces an overly restrictive inductive bias that depresses intra-class diversity and precipitates information vanishing. The effect is exacerbated in the recovery phase because the original dataset is excluded, depriving optimization of high-variance exemplars and promoting convergence to low-entropy, BN-compliant configurations rather than diverse, semantically faithful modes.

### E.2. LPLD

**To promote the intra-class diversity, LPLD** (Xiao & He, 2024) re-batches synthesis *within* each class and supervises recovery with *class-wise* BatchNorm (BN) statistics, while keeping the classification head evaluated under *global* BN for stable targets. For class $c$ with IPC images $\mathcal{C}_c = \{\tilde{x}_{c,i}\}_{i=1}^{\text{IPC}}$ and label $\tilde{y} = c$, LPLD optimizes the synthetic images via,

$$\underbrace{\mathcal{L}\Big(f_{\theta_\mathcal{O}}^{(\text{global BN})}(\tilde{x}_{c,i}),\, \tilde{y}\Big)}_{\text{classification w/ global BN}} + \alpha_{\text{BN}} \underbrace{\sum_\ell \Big\|\mu_\ell(\mathcal{C}_c) - \text{BNRM}_{\ell,c}\Big\|_2^2 + \sum_\ell \Big\|\sigma_\ell^2(\mathcal{C}_c) - \text{BNRV}_{\ell,c}\Big\|_2^2}_{\text{class-wise BN matching}}.$$

Here $\mu_\ell(\mathcal{C}_c)$ and $\sigma_\ell^2(\mathcal{C}_c)$ are the per-layer BN mean/variance computed on the *within-class* mini-batch $\mathcal{C}_c$, whereas $\text{BNRM}_{\ell,c}, \text{BNRV}_{\ell,c}$ are the *class-wise* running mean/variance obtained from $\mathcal{O}$ (squeeze stage). Their exponential moving–average (EMA) updates are

$$\text{BNRM}_{\ell,c} \leftarrow (1-\varepsilon)\,\text{BNRM}_{\ell,c} + \varepsilon\,\mu_\ell(x_c), \quad \text{BNRV}_{\ell,c} \leftarrow (1-\varepsilon)\,\text{BNRV}_{\ell,c} + \varepsilon\,\sigma_\ell^2(x_c),$$

This coupling over $\mathcal{C}_c$ enlarges intra-class diversity and improves the quality of the data.

### E.3. FADRM

FADRM (Cui et al., 2025a) synthesizes each $\tilde{x}$ by periodically fusing the *intermediate synthetic image* with a *resized real patch* from $\mathcal{O}$. Let $P_s \subset \mathcal{O}$ be the initialization patch and $D_t$ the working resolution at iteration $t$. The adjustable residual connection (ARC) applies a per-element convex fusion

$$\tilde{x}_t \leftarrow \alpha\,\tilde{x}_t + (1-\alpha)\,\text{Resample}(P_s,\, D_t), \qquad \alpha \in [0,1],$$

thereby explicitly injecting real-image content at the current resolution $D_t$ while retaining synthesized structure. Smaller $\alpha$ emphasizes high-frequency details from $P_s$; larger $\alpha$ preserves the global layout already formed in $\tilde{x}_t$. By reintroducing real-content priors along the optimization trajectory, FADRM mitigates information vanishing and yields higher-fidelity, semantically faithful synthetic data.

### E.4. RDED.

RDED (Sun et al., 2024) constructs $\mathcal{C}$ by class-preserving selection of high-confidence crops from $\mathcal{O}$. For each $(x, y) \in \mathcal{O}$, draw $K$ crops $\{x^{(k)}\}_{k=1}^K \sim \mathcal{T}(x)$ and rank them by teacher–label agreement

$$s^{(k)} = -\mathcal{L}\big(\tilde{p}(x^{(k)}),\, y\big), \qquad \tilde{p}(u) = f_{\theta_\mathcal{O}}(u).$$

Keep the image-wise best crop $x^{(\star)} = \arg\max_k s^{(k)}$, then within each class retain the top $M = N \cdot \text{IPC}$ crops for synthetic dataset construction.

## F. Implementation Details

### F.1. Datasets

We evaluate **HALD** on two benchmark datasets, **ImageNet-1K** and **Tiny-ImageNet**, both formatted as `ImageFolder`. While the original datasets contain real high-resolution natural images, our training sets are fully composed of synthetic images generated by dataset distillation methods. The validation sets remain unchanged and follow standard preprocessing pipelines.

**ImageNet-1K.** ImageNet-1K (Deng et al., 2009) contains 1,000 object classes with approximately 1.28M training images and 50K validation images. For all methods, the distilled training data are generated at resolution $224 \times 224$. During evaluation, each validation image is resized such that the shorter side is 256 pixels, followed by a center crop of size $224 \times 224$. Pixel values are normalized using the standard ImageNet statistics: mean $(0.485, 0.456, 0.406)$ and standard deviation $(0.229, 0.224, 0.225)$.

**Tiny-ImageNet.** Tiny-ImageNet (Le & Yang, 2015) is a simplified version of ImageNet with 200 classes, each having 500 training and 50 validation images. All images are pre-resized to $64 \times 64$ resolution. In our setup, distilled training images maintain this resolution. For evaluation, the validation images are directly used without additional resizing or cropping. We normalize the input images using the same statistics as ImageNet for compatibility with pretrained backbones.

### F.2. Storage Analysis

We quantify the on-disk footprint of distilled datasets under different IPC settings and compare it to the corresponding soft label storage. Despite their effectiveness, soft labels incur substantial storage overhead, often exceeding the size of the distilled images by an order of magnitude.

**Tiny-ImageNet (200 classes,** $64 \times 64$**).** As shown in Table 20, even at the lowest IPC setting (IPC=1), the original soft labels consume *over 29× more space* than the images themselves. This ratio remains consistent across IPC values due to the per-sample label overhead, leading to over 1 GB of soft labels when IPC=50, despite the images themselves occupying only 40 MB.

*Table 20.* Original soft label storage for Tiny-ImageNet.

| IPC | Image Storage | Original Soft Labels Storage |
|-----|---------------|------------------------------|
| 1   | 0.8 MB        | 23.4 MB (29.25× image storage) |
| 10  | 8 MB          | 234 MB (29.25× image storage) |
| 50  | 40 MB         | 1,170 MB (29.25× image storage) |

**ImageNet-1K (1000 classes,** $224 \times 224$**).** The storage disparity becomes even more pronounced on ImageNet-1K. As shown in Table 21, soft labels require up to *38× more storage* than images. For instance, at IPC=50, the soft labels occupy nearly **30 GB**, despite distilled images requiring less than 1 GB. Such a storage bottleneck motivates the development of more storage-efficient distillation schemes, such as partial label reuse or label reconstruction via teacher queries.

*Table 21.* Original soft label storage for ImageNet-1K.

| IPC | Image Storage | Original Soft Labels Storage |
|-----|---------------|------------------------------|
| 1   | 15 MB         | 570 MB (38× image storage) |
| 10  | 150 MB        | 5.7 GB (38× image storage) |
| 50  | 750 MB        | 28.3 GB (38× image storage) |

These results highlight that while synthetic images can be stored compactly, naive storage of soft labels becomes the primary bottleneck, especially in high-IPC or large-class-count regimes.8

### F.3. Experimental Setup

We evaluate all methods by training classification models exclusively on the distilled datasets, without any access to the original training data. Each synthetic dataset, produced by a specific distillation method, is used to supervise the training of a randomly initialized student model from scratch.

For **soft-only** baselines, the student is trained using the provided finite soft labels throughout the entire training process, with supervision applied via Kullback–Leibler (KL) divergence.

For **HALD**, we adopt a *Soft–Hard–Soft* training strategy. The model is first trained using the soft labels to leverage their fine-grained supervision. In the middle phase, hard labels are used to correct local-view semantic drift. Training then returns to soft labels in the final phase to refine the decision boundaries.

We report the validation accuracy at the final training epoch. To ensure fair and reproducible comparison, all methods are trained under an identical pipeline, with matched data augmentations, hyperparameters, and validation preprocessing steps.

### F.4. Hyper-Parameters

**Common Hyperparameters.** This part outlines the hyperparameters shared by both the *Soft-Only* baseline and **HALD**. All models are trained for 300 epochs using the AdamW optimizer with a batch size of 16. Additional details, including the learning rate and scheduler smoothing factor (denoted as Eta), are provided in Table 22 for each architecture and dataset.

*Table 22.* Hyper-parameters for all architectures on ImageNet-1K (left) and Tiny-ImageNet (right).

**ImageNet-1K (input size** $224 \times 224$**)**

| Model | IPC | Learning Rate | Eta |
|---|---|---|---|
|  | 10 | 0.0010 | 2 |
|  | 20 | 0.0010 | 2 |
| ResNet18 | 30 | 0.0010 | 2 |
|  | 40 | 0.0010 | 2 |
|  | 50 | 0.0010 | 1 |
|  | 10 | 0.0010 | 2 |
|  | 20 | 0.0010 | 2 |
| ShuffleNetV2 | 30 | 0.0010 | 2 |
|  | 40 | 0.0010 | 2 |
|  | 50 | 0.0010 | 1 |
|  | 10 | 0.0010 | 2 |
|  | 20 | 0.0010 | 2 |
| ResNet50 | 30 | 0.0010 | 2 |
|  | 40 | 0.0010 | 1 |
|  | 50 | 0.0010 | 1 |
|  | 10 | 0.0010 | 2 |
|  | 20 | 0.0010 | 2 |
| MobileNetV2 | 30 | 0.0010 | 2 |
|  | 40 | 0.0010 | 2 |
|  | 50 | 0.0010 | 2 |
|  | 10 | 0.0010 | 2 |
|  | 20 | 0.0010 | 2 |
| Densenet121 | 30 | 0.0010 | 2 |
|  | 40 | 0.0010 | 2 |
|  | 50 | 0.0010 | 2 |
|  | 10 | 0.0010 | 2 |
|  | 20 | 0.0010 | 2 |
| EfficientNet | 30 | 0.0010 | 2 |
|  | 40 | 0.0010 | 1 |
|  | 50 | 0.0010 | 1 |

**Tiny-ImageNet (input size** $64 \times 64$**)**

| Model | IPC | Learning Rate | Eta |
|---|---|---|---|
|  | 10 | 0.0010 | 2 |
|  | 20 | 0.0010 | 2 |
| ResNet18 | 30 | 0.0010 | 2 |
|  | 40 | 0.0010 | 1 |
|  | 50 | 0.0010 | 1 |
|  | 10 | 0.0010 | 2 |
|  | 20 | 0.0010 | 2 |
| ShuffleNetV2 | 30 | 0.0010 | 2 |
|  | 40 | 0.0010 | 1 |
|  | 50 | 0.0010 | 1 |
|  | 10 | 0.0010 | 2 |
|  | 20 | 0.0010 | 2 |
| ResNet50 | 30 | 0.0010 | 2 |
|  | 40 | 0.0010 | 1 |
|  | 50 | 0.0010 | 1 |
|  | 10 | 0.0010 | 2 |
|  | 20 | 0.0010 | 2 |
| MobileNetV2 | 30 | 0.0010 | 2 |
|  | 40 | 0.0010 | 1 |
|  | 50 | 0.0010 | 1 |
|  | 10 | 0.0010 | 2 |
|  | 20 | 0.0010 | 2 |
| Densenet121 | 30 | 0.0010 | 2 |
|  | 40 | 0.0010 | 1 |
|  | 50 | 0.0010 | 1 |
|  | 10 | 0.0010 | 2 |
|  | 20 | 0.0010 | 2 |
| EfficientNet | 30 | 0.0010 | 2 |
|  | 40 | 0.0010 | 1 |
|  | 50 | 0.0010 | 1 |

*Table 23.* Hard-label training duration (in epochs) for different SLC values. Left: ImageNet-1K; Right: Tiny-ImageNet.

(a) ImageNet-1K

| SLC | 300 | 250 | 200 | 150 | 100 | 50 |
|---|---|---|---|---|---|---|
| Hard Epochs | 75 | 75 | 150 | 150 | 150 | 150 |

(b) Tiny-ImageNet

| SLC | 100 | 50 |
|---|---|---|
| Hard Epochs | 50 | 50 |

**HALD-Specific Hyperparameters.** In addition to soft-label supervision, **HALD** incorporates an intermediate hard-label training phase governed by two additional hyperparameters. The first is the label smoothing rate $\alpha$, which is fixed at $0.8$ across all experiments. The second is the duration of the hard-label phase, which is aligned with the convergence time of soft-label-only training. These durations are determined empirically based on the number of soft labels available and are presented separately for ImageNet-1K and Tiny-ImageNet in Table 23, respectively.

