# OpenReview forum: "Hard Labels In! Rethinking the Role of Hard Labels in Mitigating Local Semantic Drift"
_ICML.cc/2026/Conference — ICML 2026 regular_

### Official Review · Reviewer_DUDh · 2026-03-08

**Soundness:** 4
**Presentation:** 3
**Significance:** 2
**Originality:** 2
**Overall Recommendation:** 3
**Confidence:** 3

**Summary:**

This paper investigates the local semantic drift problem in dataset distillation, where teacher-provided soft labels can become misaligned with local crops/views under limited soft-label storage. To mitigate this issue, the authors propose using hard labels as a calibration signal.

**Compliance With Llm Reviewing Policy:**

Affirmed.

**Final Justification:**

Overall, the rebuttal improves my understanding of the paper and increases my confidence in the authors' perspective, but it does not fully resolve the issues that led to my original assessment. Accordingly, I keep my score unchanged.

**Key Questions For Authors:**

Please address the concerns listed in the **Weaknesses** section.

**Limitations:**

yes

**Strengths And Weaknesses:**

## Strengths

1. The paper is well-motivated and addresses a practical bottleneck in distillation-based training under constrained soft-label storage.
2. The empirical results appear strong, especially in the low-storage regime.

------

## Weaknesses

1. **Definition of LVSD may be too weak / not tightly tied to “semantic drift.”**
    Definition 3.1 characterizes LVSD via non-zero covariance of teacher predictions across crops (i.e., Σ≠0). However, Σ≠0 does not necessarily imply “semantic drift” in the sense of drifting toward incorrect classes or conflicting with the image-level hard label. It could simply reflect reasonable uncertainty or view-dependent ambiguity. As a result, parts of the theory read more like standard statements about finite-sample estimation variance/bias, and the connection to the specific failure mode of “semantic drift” feels insufficiently tight.
2. **The claim that hard labels are content-agnostic and thus immune to drift seems overstated.**
    While hard labels do not vary with the crop, a crop may exclude the object of interest (or include mixed content, e.g., due to CutMix or aggressive cropping). Enforcing the image-level hard label on such crops can introduce label noise and potentially harm learning. The paper should more carefully justify when and why hard-label calibration improves alignment rather than injecting noise.
3. **The theoretical results do not clearly imply the optimality of a three-stage schedule.**
    It is not obvious how the provided theory leads to the conclusion that a Soft–Hard–Soft three-stage procedure is optimal (or even preferred over simpler alternatives such as a fixed mixture, adaptive weighting, or a two-stage scheme). The proposed schedule currently appears largely heuristic, and the paper would benefit from a clearer theoretical or empirical rationale isolating why three stages (and this ordering) are necessary.

------

##

---

> ### Author Rebuttal · Authors · 2026-03-31
>
> Thanks for the reviewer's careful and constructive comments, which will definitely help us improve the quality of the paper. All your suggestions will be accommodated in our revised paper. Below, we provide further detailed clarifications for each of your questions:
>
>
> > **W1: Definition of LVSD may be too weak / not tightly tied to "semantic drift".**
>
> Thanks for your kind comment. We clarify that $\Sigma\neq 0$ alone is insufficient, but the key is its interaction with the finite-$s$ regime. As suggested, we provide a tighter formalization below:
>
>  Since $\mathbb{E}[\hat{p}_s]=\bar{p}$, the supervision error w.r.t. the ground-truth label $e_y$ admits the exact identity:
>
> $$\mathbb{E}\|\hat{p}_s - e\_y\|_2^2 = \underbrace{\|\bar{p}-e\_y\|_2^2}\_{\text{(I) oracle gap, persists as }s\to\infty} + \underbrace{\frac{\operatorname{Tr}(\Sigma)}{s}}\_{\text{(II) LVSD-induced, vanishes as }s\to\infty}$$
>
> Term (I) is precisely the "benign uncertainty" the reviewer describes, i.e., genuine teacher ambiguity irreducible regardless of $s$. Term (II) is the finite-$s$ variance induced by crop-level variability. Thus, $\Sigma\neq 0$ is not itself "drift," but a source of finite-$s$ label inconsistency.
>
> To make the connection tighter, we will revise Definition 3.1 via a class-inversion event: for $c\neq y$,
>
> $$\mathcal{E}_{s,c} := {\hat{p}_s(c) \geq \hat{p}_s(y)}.$$
>
> If $\bar{p}_y > \bar{p}c$, define $D{s,c} := \hat{p}_s(y) - \hat{p}_s(c)$, which satisfies
>
> $$\operatorname{Var}(D_{s,c}) = \frac{\Sigma_{yy} + \Sigma_{cc} - 2\Sigma_{yc}}{s}.$$
>
> Applying the one-sided Cantelli inequality with $t = \bar{p}_y - \bar{p}_c > 0$:
>
> $$\Pr(\mathcal{E}{s,c}) \leq \frac{(\Sigma{yy}+\Sigma_{cc}-2\Sigma_{yc})/s}{(\Sigma_{yy}+\Sigma_{cc}-2\Sigma_{yc})/s + (\bar{p}_y - \bar{p}_c)^2}.$$
>
> This bound is distribution-free, monotonically decreasing in $s$, and vanishes as $s\to\infty$. We will revise the paper to make explicit that LVSD refers to this finite-$s$ risk of class inversion, rather than to $\Sigma\neq 0$ alone.
>
>
> > **W2: The claim that hard labels are content-agnostic and thus immune to drift seems overstated.**
>
> Thanks for raising this point. We clarify that our claim is that hard labels provide *a fixed global semantic anchor* across crops, unlike crop-dependent soft predictions, not that they are completely noise-free. Crops without clear object evidence can introduce weak/noisy supervision, but they are not always pure noise, since background may still provide useful context, such as water for fish or grass for cows. Empirically, background-only crops perform worst, while combining object and contextual regions gives the best results.
>
> To justify this, we conducted a controlled Stage-B experiment below:
>
> | Background-only | Object-only | HALD |
> |:----:|:----:|:----:|
> | 27.9 | 35.3 | **35.6** |
>
> The background-only setting performs worse as expected, since such crops are usually less informative and uncommon. However, our HALD matches or slightly surpasses the object-only variant, suggesting that contextually relevant crops dominate in practice and that occasional uninformative crops do not harm learning. We will revise the relevant part to make clear that hard-label calibration reduces crop-induced semantic drift on average, while still acknowledging that aggressive background-only or mixed-content crops may introduce some noise.
>
> > **W3: The theoretical results do not clearly imply the optimality of a three-stage schedule.**
>
> Thanks for the comment. We clarify that our current theorems indeed can imply a simple thresholded control model under which the optimal binary schedule is exactly *Soft–Hard–Soft*. First we define the terminal objective $J_T = b_T + \lambda v_T$, where from our exact bias-variance decomposition:
>
> $$\mathbb{E}\|\hat{p}_s - e_y\|_2^2 = \underbrace{\|\bar{p}-e_y\|_2^2}\_{b_t \text{ (oracle gap)}} + \underbrace{\frac{\operatorname{Tr}(\Sigma)}{s}}\_{v_t \text{ (LVSD variance)}}$$
>
> Let $u_t \in \{S, H\}$ and define alignment $\rho_t$. Suppose: (i) $S$ increases $\rho_t$ until threshold $\rho_c$; (ii) $H$ reduces $v_t$ via Corollary 1, i.e., $v_t \to \operatorname{Tr}(\Sigma)/s_\text{eff}(t)$, only when $\rho_t \geq \rho_c$, and only until $v_t$ reaches floor $v_c$; (iii) $S$ reduces $b_t$ only after $v_t \leq v_c$.
>
> **Optimality argument.** Three dominance statements follow directly:
>
> - **Before $\rho_t \geq \rho_c$:** $H$ is ineffective ($s_\text{eff} \approx s$, $v_t$ unchanged), so $S$ strictly dominates.
> - **After $\rho_t \geq \rho_c$, before $v_t \leq v_c$:** $H$ strictly reduces $v_t$ while $S$ cannot, so $H$ strictly dominates.
> - **After $v_t \leq v_c$:** $v_t$ is saturated and $H$ no longer reduces $J_T$, so $S$ strictly dominates by reducing $b_t$.
>
> Hence any optimal binary schedule must have the form $S^{T_A}H^{T_B}S^{T_C}$. This abstraction is directly grounded in Theorem 3.7, Corollary 3.8, and the bias-variance decomposition, and is consistent with Tables 11 and 15.

---

> > ### Author Rebuttal · Reviewer_DUDh · 2026-04-04
> >
> > Thanks to the authors for the detailed rebuttal. The response addresses my concerns constructively and improves my confidence in the paper. In particular, the clarification that $\Sigma \neq 0$ is not itself the semantic drift but rather a source of finite-sss supervision inconsistency makes the LVSD discussion more precise. I also appreciate the toned-down claim regarding hard labels: the rebuttal now frames them as a global semantic anchor rather than noise-free supervision, which is a more defensible position. Finally, while I still think the theoretical results do not fully establish the optimality of the Soft–Hard–Soft schedule, the authors provide a clearer rationale and some additional empirical support for why this ordering is preferred.
> >
> > Overall, my main concerns are only partially resolved rather than fully eliminated, especially regarding the strength of the theoretical justification for the three-stage schedule. However, I find the rebuttal sufficiently helpful to improve my assessment of the work. I will update my score accordingly.

---

> > > ### Author Response · Authors · 2026-04-05
> > >
> > > Thanks for your kind and encouraging follow-up comments. Our current theory provides a good starting point. As the reviewer also mentioned, although it still has room to improve, we are confident that it offers a meaningful and important step forward for this direction and for understanding this problem more broadly. We also appreciate your thoughtful assessment. We are glad that the rebuttal helped clarify the role of LVSD and our use of hard labels as global semantic anchors rather than noise-free supervision. In the revision, we will further calibrate the claims appropriately and strengthen the presentation of both the rationale and the empirical support, we will also incorporate all these additional refined theory in the rebuttal, more results, and clarifications into the revised paper. Thank you again for your constructive comments and for updating your assessment of the paper. Please feel free to let us know if you have any further questions.

---

### Official Review · Reviewer_DBaU · 2026-03-12

**Soundness:** 2
**Presentation:** 3
**Significance:** 3
**Originality:** 3
**Overall Recommendation:** 5
**Confidence:** 3

**Summary:**

This paper proposes to utilize hard labels to both address the massive storage bottleneck and improve performance for dataset distillation methods. The paper identifies that reducing cropped views introduce a problem called local-view semantic drift, when crops are assigned soft labels that deviate from the ground-truth image label. To solve this, the paper proposes a 3-stage training scheme that consists of 2 soft-label stages and a key hard-label stage in the middle that theoretically improve the gradient alignment of the two label types. Massive theoretical analysis and a semantic drift framework are provided to support the method. Experiments demonstrate state-of-the-art performance and a 100x reduction of storage cost.

**Compliance With Llm Reviewing Policy:**

Affirmed.

**Final Justification:**

The authors have extensively addressed the concerns regarding both theoretical validity and empirical strength. With the inclusion of these new results and clarifications, the paper is now comprehensive and well-supported. Therefore, I would raise my recommendation to 5: accept.

**Key Questions For Authors:**

1. Why is the one-hot labels not used for stage B directly?
2. How is the LS + Cutmix method able to perform gradient correction?
3. Is there intuitive explanation or empirical evidence on how the three stages work to bound the gap in Theorem 3.5?

**Limitations:**

Effectiveness on other benchmarks are currently not clear.

**Strengths And Weaknesses:**

Strengths:

The theoretical framework of the soft-hard alignment is thorough in this paper, with in-depth analysis of the Local-View Semantic Drift problem studied in the paper. Meanwhile, the empirical validation shows strong effectiveness and efficiency.

Weaknesses:
- The main method, i.e., training stage B, seems not using hard labels for correcting signals, which is the primary motivation of the paper. The method basically uses label smoothing + Cutmix, both still creating soft labels.
- The theoretical part before Sec.3.2 is quite disconnected to the main method.
- The number of compared methods are limited. The authors are suggested to compare 1~2 more (e.g., GIFT).
- The evaluated datasets are limited to only Tiny-ImageNet/ImageNet-1K, limiting it only in general domains.
- Minor issues:
  - Notation of $s$ defined for the second time at page 4 stage A.
  - Table 6 caption: right/left rather than top/bottom.
  - The best-performing $\alpha$ is annotated wrong in Table 13.

---

> ### Author Rebuttal · Authors · 2026-03-31
>
> We thank the reviewer for the detailed and constructive feedback, which will definitely help improve the paper. We will accommodate all suggestions in the revision. Below, we provide point-by-point clarifications to each question:
>
> > **W1: The main method training stage B not using hard label.**
>
> Thanks for the comment. In our paper, "hard label" refers to the *global image-level anchor label*, in contrast to teacher-produced adaptive soft labels, rather than to a strictly one-hot target. The key distinction is **content adaptivity**: teacher soft labels vary with crop content, whereas Stage B uses a fixed rule anchored to the global class $y$. In this sense, Stage B targets are better viewed as **heavily flattened hard labels**. Since our data are synthetic and less semantically certain than real data, stronger flattening is needed for stable calibration, which is also supported by our results below:
>
> | | $\alpha$=0.0 | $\alpha$=0.8 |
> |:--:|:--:|:--:|
> | Acc (%) | 35.3 | **35.6** |
>
> > **W2: Theory-method connection.**
>
> As suggested, we provide a tighter LVSD definition that directly links the theory to the method through $s\_\text{eff}$.
>
> **Tighter Definition of LVSD.** Since $\mathbb{E}[\hat{p}\_s]=\bar{p}$, the supervision error admits the exact identity:
> $$\mathbb{E}\|\hat{p}\_s - e\_y\|\_2^2 = \underbrace{\|\bar{p}-e\_y\|\_2^2}\_{\text{(I) oracle gap}} + \underbrace{\frac{\operatorname{Tr}(\Sigma)}{s}}\_{\text{(II) LVSD-induced variance}}$$
> Term (I) is irreducible; Term (II) is the finite-$s$ artifact HALD directly targets. To connect this to class-level semantic errors, we define a **class-inversion event**: for $c\neq y$,
> $$\mathcal{E}\_{s,c} := \{\hat{p}\_s(c) \geq \hat{p}\_s(y)\}.$$
> Applying the one-sided Cantelli inequality with $t = \bar{p}\_y - \bar{p}\_c > 0$:
> $$\Pr(\mathcal{E}\_{s,c}) \leq \frac{(\Sigma\_{yy}+\Sigma\_{cc}-2\Sigma\_{yc})/s}{(\Sigma\_{yy}+\Sigma\_{cc}-2\Sigma\_{yc})/s + (\bar{p}\_y - \bar{p}\_c)^2}.$$
> This bound is distribution-free, monotonically decreasing in $s$, and vanishes as $s\to\infty$.
>
> **Direct Connection to HALD.** By Corollary 3.8, Stage B increases $s$ to $s\_\text{eff} = s/(1-\rho_*^2)$, directly tightening the Cantelli bound: $\Pr(\mathcal{E}\_{s\_\text{eff},c}) < \Pr(\mathcal{E}\_{s,c})$. This closes the loop between Sec. 3.1 and 3.2, HALD reduces class-inversion risk by increasing $s\_\text{eff}$ via hard-label calibration. We will revise accordingly.
>
> > **W3: The number of compared methods.**
>
> Following your suggestion, we add a more comparison with LPQLD under a fixed storage budget of SLC=300 (570 MB), using LPLD-generated images for all methods. HALD consistently outperforms all baselines in this setting.
>
> |IPC| LPLD | LPQLD | HALD (Ours) |
> |:-:|:--:|:--:|:--:|
> | IPC=10 |32.7| 32.8 | **36.2** |
> | IPC=20 |41.0| 42.3 | **43.9** |
>
> We also kindly note that GIFT and several other baselines are already included in our original submission (Table 3).
>
> > **W4: Evaluated datasets beyond Tiny-ImageNet/ImageNet-1K.**
>
> Thanks for the suggestion. Following your comment, we additionally evaluate HALD on CIFAR-100 and ImageWoof datasets under IPC=10, where it consistently outperforms the soft-only baseline:
>
> |Dataset|SLC|Soft-Only | HALD |
> |:--:|:--:|:--:|:--:|
> | CIFAR-100| 10 (280 KB)|10.6 |   **21.0** (+10.4) |
> | CIFAR-100| 20 (560 KB)|16.5 | **26.0** (+9.5) |
> | ImageWoof| 10 (24 KB)| 23.5 |  **27.4** (+3.9)  |
> | ImageWoof| 20 (48 KB)| 27.4 |  **29.3** (+1.9) |
>
> > **W5: Minor issues / typos.**
>
> Thanks for pointing them out. We will carefully correct them in the revision and thoroughly revise the whole paper again.
>
> > **Q1: Why are the one-hot labels not used for stage B directly?**
>
> Stage B labels are better viewed as **heavily flattened hard labels**, we will revise this in revision. Please also see our response to **W1** for details.
>
> > **Q2: How is the LS + Cutmix method able to perform gradient correction?**
>
> The key is that $t_{\lambda,\alpha}(y, y’)$ depends only on the ground-truth labels of the two mixed images, not on local crop content. This fixed global anchor makes Stage B corrective: unlike Stage A's crop-dependent soft labels, which are prone to semantic drift, Stage B always pulls toward the true class anchor regardless of crop content. Table 6 (left) supports this, where crop consistency increases from 0.74 to 0.96 after Stage B.
>
> > **Q3: Intuitive or empirical evidence to bound the gap in Theorem 3.5.**
>
> Thanks for the question. Theorem 3.5 shows that the supervision gap to full soft-label coverage scales as $O(1/\sqrt{s})$, so the direct remedy is to increase $s$. But storing more soft labels is exactly the bottleneck we target. HALD avoids this extra cost: by Corollary 3.8, Stage B increases the effective sample size to $s_\text{eff}=s/(1-\rho_*^2)>s$, yielding a strictly tighter bound. Figure 2 and Table 6 (right) support this, with prediction consistency to a full soft-label coverage reference model improving from 0.46 to 0.62 with hard calibration.

---

> > ### Author Rebuttal · Reviewer_DBaU · 2026-04-01
> >
> > The authors have addressed my concerns in detail for theoretical validity and empirical strength. I'll raise my score accordingly.

---

> > > ### Author Response · Authors · 2026-04-03
> > >
> > > Thanks and we are very glad to hear this. We will incorporate all these additional experiments, results, and clarifications into the revised paper. Thank you again for your constructive comments and please feel free to let us know if you have any further questions.

---

### Official Review · Reviewer_yuXU · 2026-03-12

**Soundness:** 3
**Presentation:** 3
**Significance:** 3
**Originality:** 3
**Overall Recommendation:** 5
**Confidence:** 4

**Summary:**

This paper addresses an important efficiency challenge in large-scale dataset distillation: the storage overhead of precomputed soft labels, referred to as Soft Label Capacity (SLC). The authors identify a phenomenon termed Local Semantic Drift (LVSD), where limiting the number of crops per image causes soft supervision to deviate from the global ground-truth semantics due to visual ambiguity in individual crops.
To mitigate this issue, the paper proposes HALD, a Soft–Hard–Soft (S-H-S) training paradigm in which hard labels are introduced in the middle phase as semantic anchors to recalibrate the learning process. The paper also provides a theoretical analysis of the objective mismatch under limited soft-label supervision and demonstrates the effectiveness of HALD on Tiny-ImageNet and ImageNet-1K.

**Compliance With Llm Reviewing Policy:**

Affirmed.

**Final Justification:**

The rebuttal effectively addressed my main concerns by introducing an adaptive phase-switching mechanism and providing additional results on fine-grained benchmarks (e.g. ImageWoof). These clarifications significantly improve the practical credibility of the method. The remaining discussion regarding the "hard label" terminology appears to be largely a matter of interpretation and does not undermine the methodological validity or empirical effectiveness of HALD.

**Key Questions For Authors:**

1. How does the hard-label calibration phase behave when crops contain little or no object-related information (e.g., background-only crops)?

2. Is there a heuristic or adaptive strategy to determine the phase durations (η) for new datasets without extensive hyperparameter tuning?

3. Do the authors expect the Soft–Hard–Soft schedule to remain effective for fine-grained classification tasks, where local crops across classes may be visually similar?

4. For Table 3, were the baseline methods re-trained under the same SLC constraints and augmentation settings, or were the reported numbers taken from the original papers?

5. Given that the optimal Label Smoothing rate is α=0.8 (leaving only 20% weight for the ground-truth class), how do the authors justify the terminology "Hard Label" for the Stage B calibration phase? Could the observed performance gains be primarily attributed to strong uniform regularization rather than the semantic anchoring of a hard label?

**Limitations:**

yes

**Strengths And Weaknesses:**

**Strengths**
1. **Practical motivation**:
Reducing the storage cost of soft labels is a highly relevant problem for scaling dataset distillation to real-world datasets. The focus on SLC provides a clear and practical motivation.

2. **Conceptual perspective**:
Revisiting the role of hard labels in modern distillation pipelines is an interesting perspective. Treating hard labels as semantic anchors that correct drifted soft supervision provides a clear and intuitive interpretation.

3. **Empirical performance**:
HALD consistently improves performance across multiple benchmarks, particularly in storage-constrained regimes. The gains over established baselines such as SRe2L and LPLD appear meaningful.

**Weaknesses**
1. **Robustness of the Semantic Anchor Mechanism**:
The paper argues that hard labels serve as content-agnostic semantic anchors that correct Local Semantic Drift. However, it remains somewhat unclear how this mechanism behaves when crops contain little or no object-related information.
Prior work such as ReLabel suggests that assigning global labels to arbitrary crops may introduce supervision noise when the crop mainly contains background. It would be helpful if the authors could discuss how the calibration phase behaves in such cases, and whether background-heavy crops could affect the stability of the proposed method.

2. **Sensitivity to the Phase Scheduling (η)**:
The effectiveness of HALD appears to depend on the duration of the Soft–Hard–Soft phases. As shown in Table 12, performance varies depending on the chosen scheduling.
While the 75/75 split works well for the reported benchmarks, the paper does not provide clear guidance on how these phase durations should be selected for new datasets or tasks. This may become particularly relevant for tasks such as fine-grained classification, where the characteristics of semantic drift may differ.
A brief discussion of possible heuristics or adaptive strategies for determining the phase transitions could improve the practical usability of the method.

3. **Interpretation of Gradient Cosine Similarity**:
In Section 3.2, the paper motivates the phased training strategy by observing low cosine similarity between soft-label and hard-label gradients during early training.
While this observation is interesting, it is not entirely clear why low similarity should necessarily be interpreted as harmful interference. In many optimization settings, gradients from different objectives can provide complementary information. A more detailed explanation of why joint optimization leads to degraded performance in this case would help clarify the motivation for the phased design.

4. **Clarity of Experimental Setup in Table 3**:
The paper emphasizes fair comparisons under identical storage budgets. However, the presentation in Table 3 is somewhat less transparent than in Table 1.
Table 3 mainly reports results as a function of SLC, but it does not clearly indicate the corresponding IPC (Images Per Class) settings for all baselines. Providing a clearer description of the experimental configuration would make the comparisons easier to interpret and improve reproducibility.

5. **Contradiction in the "Hard Label" Terminology and Excessive Label Smoothing**: The paper's title and core motivation revolve around using "Hard Labels" as semantic anchors. However, in Stage B, the calibration loss uses Label Smoothing LS_α(y)=(1−α)*δ_y+α*1/C. According to the ablation study in Table 13, the optimal performance is consistently achieved at an extremely high smoothing rate of α=0.8. This implies that the actual ground-truth label retains only 20% of the weight, while the remaining 80% is distributed uniformly across all classes. It is highly questionable whether a target heavily dominated by an 80% uniform distribution can still be defined as a "Hard Label." This mechanism appears to act more like heavy uniform regularization rather than a strict hard-label anchor, which seems to contradict the main narrative and title of the paper.

---

> ### Author Rebuttal · Authors · 2026-03-31
>
> Thanks for the thorough and constructive comments, which will definitely help us improve the quality of the paper. We will carefully accommodate all your suggestions in our revision. Below, we provide detailed clarifications for each of your questions:
>
> > **W1: Robustness of the Semantic Anchor Mechanism.**
>
> Thanks for the insightful comment. We clarify that background is not always pure noise, it can provide useful object context. For instance, water-only crops may still be informative for fish because water and fish strongly co-occur. More generally, our semantic anchor is meant to reduce local semantic drift at the image level, not to assume every crop is independently object-discriminative. Thus, while some background-only crops are weakly informative, many still provide contextual cues that help stabilize semantic consistency. We will clarify this in the revision.
>
> > **W2: Sensitivity to Phase Scheduling (η).**
>
> Thanks for the comment. We clarify that our phase scheduling can be easily applied to new datasets via an *adaptive phase-switching mechanism* with two theory-guided proxies:
> - **Phase 0→1 (Soft→Hard):** triggers when validation accuracy growth becomes marginal, indicating stabilized soft/hard gradient alignment;
> - **Phase 1→2 (Hard→Soft):** triggers when normalized student entropy $H_t/\log C$ plateaus, indicating hard calibration saturation, also consistent with Corollary 3.8.
>
> The $\log C$ normalization makes the trigger comparable across datasets. As shown below, this adaptive schedule matches the fixed one across both standard and fine-grained settings, avoiding manual tuning.
>
> | Dataset | HALD (adaptive) | HALD (fixed) |
> |:----:|:----:|:----:|
> | CIFAR-100  | 25.8| 26.0|
> | ImageWoof | 27.5 |27.4 |
> | ImageNet-1K | 35.2 | 35.6 |
>
> We will make this clearer in revision.
>
> > **W3: Interpretation of Gradient Cosine Similarity.**
>
> Thanks for the comment. Our claim is not that low cosine similarity is always harmful, but that in our setting strong early misalignment makes joint optimization less effective. As Fig. 3 (left) shows, the gradients are far apart early on but align later, suggesting hard labels are mainly useful in the early and middle stages, while soft labels are sufficient in the final stage. This motivates our phased design in the paper.
>
> > **W4: Clarity of Experimental Setup in Table 3.**
>
> All results in Table 3 use IPC=30 with identical augmentation and hyperparameter settings, ensuring a fair comparison under the same storage budget. We will clarify this in revision.
>
> > **W5: "Hard Label" Terminology and Excessive Label Smoothing.**
>
> Thanks for the comment. In our paper, "hard label" refers to the global image-level anchor label in contrast to teacher-produced adaptive soft labels, not to a strictly one-hot target. Stage B smoothing is a fixed rule applied uniformly to all samples, so it remains anchored to the same global class label even under strong smoothing, unlike teacher soft labels, which vary across instances.
> In the revision, we will clarify that these are better described as *heavily flattened hard labels*. Since our synthetic data has less clear semantics than real data, stronger flattening is needed for stable calibration, as also supported by our one-hot comparison below:
>
> |  | $\alpha$=0.0 | $\alpha$=0.8 |
> |:-:|:-:|:-:|
> | Acc (%) | 35.3 |  35.6 |
>
> > **Q1: Hard-label calibration phase behave when background-only crops.**
>
> As discussed in **W1**, background-only crops can still be informative. Empirically, background-only performs worst, object-only is also suboptimal, and the best results come from using both object and contextual regions. This suggests informative contextual crops dominate, while occasional uninformative ones do not harm training much.
>
> | Background-only | Object-only | HALD |
> |:-:|:-:|:-:|
> | 27.9 | 35.3 | **35.6** |
>
> > **Q2: Is there a heuristic or adaptive strategy.**
>
> Yes, we propose an adaptive phase-switching mechanism based on validation accuracy and normalized student predictive entropy (see **W2** for more details).
>
> > **Q3: Will soft–Hard–Soft remain effective for fine-grained tasks?**
>
> Following your suggestion, we evaluate HALD on ImageWoof, a fine-grained dog-breed benchmark, and results below consistently outperform the soft-only baseline, confirming its effectiveness in fine-grained settings.
>
> |Dataset|SLC|Soft-Only | HALD |
> |:-:|:-:|:-:|:-:|
> | ImageWoof| 10 (24 KB)| 23.5 |  **27.4** (+3.9)  |
> | ImageWoof| 20 (48 KB)| 27.4 |  **29.3** (+1.9)  |
>
> > **Q4: Clarifications of baseline methods re-training settings in Table 3.**
>
> Yes, we confirm that all baseline methods are re-trained under identical SLC constraints and augmentation settings.
>
> > **Q5: How do the authors justify the terminology "Hard Label".**
>
> Thanks. We clarify that our "hard label" refers to the **global image-level anchor label**, in contrast to the teacher-produced adaptive soft label, rather than to a strictly one-hot target.

---

> > ### Author Rebuttal · Reviewer_yuXU · 2026-04-03
> >
> > I thank the authors for their detailed response. My concerns regarding the adaptive scheduling and the effectiveness in fine-grained classification have been well addressed by the additional results and explanations, and I will consider raising my score accordingly.

---

> > > ### Author Response · Authors · 2026-04-03
> > >
> > > Thanks and we are very glad to hear this. We will incorporate all these additional experiments, results, and clarifications into the revised paper. Thank you again for your constructive comments and please feel free to let us know if you have any further questions.

---

### Official Review · Reviewer_YkdD · 2026-04-03

**Soundness:** 2
**Presentation:** 2
**Significance:** 2
**Originality:** 2
**Overall Recommendation:** 3
**Confidence:** 2

**Summary:**

This paper revisits the overlooked role of hard labels and proposes a new method named HALD, which uses hard labels as intermediate corrective signals while preserving the fine-grained benefits of soft labels. The proposed method achieves 42.7% accuracy with only 285M soft-label storage.

**Compliance With Llm Reviewing Policy:**

Affirmed.

**Final Justification:**

After reading the rebuttal and the comments from the other reviewers, I decided to maintain my score and believe the paper is below the bar of ICML.

**Key Questions For Authors:**

See above.

**Limitations:**

yes

**Strengths And Weaknesses:**

Pros.
- The problem studied in this paper is both interesting and significant.
- The paper is clearly written and well-structured.

Cons.
- More case studies, such as visualization, should be included to demonstrate the effectiveness of the proposed method.
- More generation methods should be included in the performance comparison. The newest compared method is not published. I suggest authors choose more baselines in published papers for a more solid comparison.
- The problem is not clear. I suggest the authors include a clear problem definition in the title.
- The baseline seems to be very weak, which is much weaker than model variants in the following experiments.
- The novelty of the proposed method seems to be carefully illustrated since hard labels and soft labels are very popular in  knowledge transfer and large-scale dataset distillation.

---

### Decision · Program_Chairs · 2026-04-30

**Decision:**

Accept (regular)

**Comment:**

The paper proposes to reconsider the role of hard labels in knowledge distillation as they can serve as a semantic anchor to correct the drift arising in soft-label-based approaches for knowledge distillation when using a limited number of image crops. The reviewers requested some clarifications about the problem formulation and other aspects (e.g. Gradient Cosine Similarity) and to evaluate the method in other settings. The authors provided several additional results and clarifications about the theoretical validity of the proposed approach. Overall, most of the reviewers found that the most important concerns were addressed in the rebuttal. However, considering the overall feedback from the reviewers and the overlap in the concerns among different reviews, the AC recommends acceptance.